# The acetylase activity of Cdu1 regulates bacterial exit from infected cells by protecting *Chlamydia* effectors from degradation

**Robert J Bastidas[1]\*, Mateusz Kędzior[1], Robert K Davidson[2†], Stephen C Walsh[2†], Lee Dolat[1], Barbara S Sixt[3,4,5], Jonathan N Pruneda[6], Jorn Coers[1,2], Raphael H Valdivia[1,2]**

[1]Department of Integrative Immunobiology, Duke University, Durham, United States; [2]Department of Molecular Genetics and Microbiology, Duke University, Duke, United States; [3]Deparment of Molecular Biology, Umeå University, Umeå, Sweden; [4]The Laboratory for Molecular Infection Medicine Sweden (MIMS), Umeå University, Umeå, Sweden; [5]Umeå Centre for Microbial Research (UCMR), Umeå University, Umeå, Sweden; [6]Department of Molecular Microbiology & Immunology, Oregon Health & Science University, Portland, United States

**\*For correspondence:**
robert.bastidas@duke.edu

†These authors contributed equally to this work

**Competing interest:** The authors declare that no competing interests exist.

**Abstract** Many cellular processes are regulated by ubiquitin-mediated proteasomal degradation. Pathogens can regulate eukaryotic proteolysis through the delivery of proteins with de-ubiquitinating (DUB) activities. The obligate intracellular pathogen *Chlamydia trachomatis* secretes Cdu1 (ChlaDUB1), a dual deubiquitinase and Lys-acetyltransferase, that promotes Golgi remodeling and survival of infected host cells presumably by regulating the ubiquitination of host and bacterial proteins. Here, we determined that Cdu1's acetylase but not its DUB activity is important to protect Cdu1 from ubiquitin-mediated degradation. We further identified three *C. trachomatis* proteins on the pathogen-containing vacuole (InaC, IpaM, and CTL0480) that required Cdu1's acetylase activity for protection from degradation and determined that Cdu1 and these Cdu1-protected proteins are required for optimal egress of *Chlamydia* from host cells. These findings highlight a non-canonical mechanism of pathogen-mediated protection of virulence factors from degradation after their delivery into host cells and the coordinated regulation of secreted effector proteins.

## eLife assessment

This **important** study combines state-of-the art proteomics and genetic manipulation of *Chlamydia trachomatis* to study the function of a chlamydial effector, Cdu1, with deubiquitination and acetylation activities. **Solid** evidence is provided to show that Cdu1 is able to protect itself and three other chlamydial effectors, which are involved in the control of chlamydial egress from host cells, from ubiquitin-mediated degradation, and that this depends on the acetylation activity of Cdu1, but not on its deubiquitination activity. This work will be of interest to microbiologists and cell biologists studying host cell-pathogen interactions.

## Introduction

Ubiquitination is a conserved and ubiquitous post-translational modification (PTM) of proteins involving the conjugation of the carboxy-terminal glycine residue of ubiquitin (Ub) to lysine residues

of target proteins. Poly-ubiquitination of substrates involves further conjugation of a Ub internal lysine residue or amino-terminal methionine (M1) with a second Ub molecule. Seven internal lysines in Ub (K6, K11, K27, K29, K33, K48, K63) and M1 are utilized by Ub conjugating enzymes to form homogeneous, branched, or mixed poly-ubiquitin (polyUb) chains (*Komander and Rape, 2012*). PolyUb chains with different linkage types exhibit distinct structures and functions. For example, K48- and K11-linked polyUb chains exhibit a compact conformation and are substrates for 26 S proteasome-mediated degradation (*Varadan et al., 2002*; *Tenno et al., 2004*; *Eddins et al., 2007*; *Bremm et al., 2010*; *Saeki, 2017*). In contrast, K63-linked polyUb conjugates adopt more open conformations that enable the recruitment of multiprotein complexes that regulate the function of the target protein by proteolytic independent events (*Komander et al., 2009b*; *Weeks et al., 2009*; *Datta et al., 2009*; *Komander and Rape, 2012*). Mixed and branched polyUb chains are also emerging as important regulators of physiological functions (*Swatek and Komander, 2016*; *Ohtake and Tsuchiya, 2017*).

Protein ubiquitination regulates numerous eukaryotic cell processes including protein degradation, signal transduction, cell cycle regulation, selective autophagy, the DNA damage response, and programmed cell death. Ub also plays key roles in modulating host innate immune responses to bacterial infection (*Li et al., 2016*), bacterial proteins, pathogen-containing vacuoles, and bacteria themselves by targeting them for Ub-mediated degradation by proteasomal or autophagic machineries (*Li et al., 2016*). Because the Ub system is critical for pathogen containment, many pathogens have evolved mechanisms to counteract the impact of this PTM (*Vozandychova et al., 2021*). For instance, bacterial deubiquitinases (DUBs) can remove Ub from ubiquitinated substrates thereby dampening inflammatory and cell-autonomous defense mechanisms (*Kubori et al., 2019*). Many DUBs are cysteine proteases with a catalytic Cys, a nearby His and an Asn/Asp (*Komander et al., 2009a*). DUBs are typically dedicated to the removal of Ub moieties and are unable to hydrolyze other Ub-like (Ubl) modifications such as SUMO or NEDD8. However, the CE clan of Ubl proteases (ULPs) can catalyze the removal of both SUMO and NEDD8 (*Ronau et al., 2016*). Bacterial pathogens also encode CE clan enzymes that function as DUBs, ULPs or both. For instance, *Salmonella* Typhimurium SseL, *Escherichia coli* ElaD, and *Shigella flexneri* ShiCE function as Ub specific proteases (*Rytkönen et al., 2007*; *Catic et al., 2007*; *Pruneda et al., 2016*) while RickCE from *Rickettsia belli* functions as a protease directed towards both Ub and NEDD8 as does SidE from *Legionella pneumophila* which displays mixed activities towards Ub, NEDD8, and ISG15 (*Sheedlo et al., 2015*; *Pruneda et al., 2016*). Similarly, XopD from *Xanthamonas campestris* and LotB from *L. pneumophila* are isopeptidases exhibiting cross reactivity towards both Ub and SUMO (*Pruneda et al., 2016*; *Schubert et al., 2020*). Some CE clan bacterial effectors display acetyltransferase activity. *L. pneumophila* LegCE, *S.* Typhimurium AvrA, and YopJ from *Yersinia pestis* function exclusively as acetyltransferases (*Mittal et al., 2006*; *Mukherjee et al., 2006*; *Jones et al., 2008*; *Pruneda et al., 2016*). In contrast, the *Chlamydia trachomatis* (Ct) effector Cdu1/*Chla*DUB1 is a CE clan protein that exhibits both acetyl-transferase and deubiquitinating activities (*Misaghi et al., 2006*; *Pruneda et al., 2016*; *Fischer et al., 2017*; *Pruneda et al., 2018*).

Ct is an obligate intracellular bacterial pathogen responsible for human diseases of significant clinical and public health importance (*Haggerty et al., 2010*). Ct has a biphasic developmental cycle in which the Ct infectious propagule or elementary body (EB) invades the target host cell. Upon internalization the EB transitions to the reticulate body (RB). RBs replicate by binary fission within a pathogenic vacuole ('inclusion') and asynchronously differentiate back to EBs. In cell culture, starting at around 48 hours-post infection (hpi) Ct will egress after lysis of the host cell or by a process termed extrusion, wherein the intact inclusion exits from the infected cell (*Moulder, 1991*; *Abdelrahman and Belland, 2005*; *Hybiske and Stephens, 2007*; *Lee et al., 2018*). The effector Cdu1 was originally identified as a deneddylating and deubiquitinating enzyme and subsequently shown to exhibit in vitro isopeptidase activity towards both Lys48 and Lys63 linked di-Ub substrates (*Misaghi et al., 2006*; *Claessen et al., 2013*; *Pruneda et al., 2016*; *Fischer et al., 2017*). Cdu1 is unique among CE clan enzymes in that it also functions as a bona fide lysine acetylase with both is acetylase and DUB activities catalyzed by the same catalytic active site (*Pruneda et al., 2018*). Intriguingly, Cdu1 autoacetylation is directed towards lysines unlike other CE clan acetylases that predominantly target serine and threonine residues (*Pruneda et al., 2018*). In transfected cells, Cdu1 protects the NFκB cytoplasmic retention factor IκBα from ubiquitination and proteasomal degradation (*Le Negrate et al., 2008*). In infected cells, Cdu1 localizes to the inclusion membrane where it functions to stabilize the anti-apoptotic protein

Mcl-1 and to promote the repositioning of Golgi ministacks around the Ct inclusion (*Fischer et al., 2017*; *Wang et al., 2018*; *Pruneda et al., 2018*; *Kunz et al., 2019*; *Auer et al., 2020*). However, the mechanism by which Cdu1 promotes redeployment of Golgi ministacks and any additional roles that Cdu1 may play during Ct infection of epithelial cells remains unknown.

In this study, we show that Cdu1 protects itself and three secreted Ct effectors, InaC, IpaM, and CTL0480 from targeted ubiquitination and proteasomal degradation. InaC, IpaM, and CTL0480 are members of a larger family of bacterial proteins embedded within the inclusion membrane (Inc proteins; *Bannantine et al., 2000*; *Rockey et al., 2002*; *Chen et al., 2006*; *Li et al., 2008*; *Alzhanov et al., 2009*; *Dehoux et al., 2011*; *Lutter et al., 2012*; *Lutter et al., 2013*; *Kokes et al., 2015*; *Weber et al., 2015*). We show that Cdu1-mediated protection from degradation is independent from its DUB activity but relies upon its Lys acetylase activity. We show that Cdu1 protects InaC to promote repositioning of Golgi ministacks and formation of actin scaffolds around the Ct inclusion, and CTL0480 to promote recruitment of myosin phosphatase target subunit 1 (MYPT1) to the inclusion. In addition, we determined that Cdu1 and Cdu1-protected Incs are required for optimal extrusion of inclusions from host cells at the late stages of infection.

## Results

### The *C. trachomatis* inclusion membrane proteins InaC, IpaM, and CTL0480 are differentially ubiquitinated in the absence of Cdu1

Cdu1 is required for Golgi repositioning around the Ct inclusion (*Pruneda et al., 2018*; *Auer et al., 2020*). To understand how Cdu1 promotes Golgi redistribution, we first generated a *cdu1* null strain in a Ct L2 background by TargeTron mediated insertional mutagenesis (pDFTT3-*aadA*) (*Lowden et al., 2015*; *Figure 1—figure supplement 1*). Loss of Cdu1 expression in the resulting L2 *cdu1*::GII *aadA* (*cdu1*::GII) strain was verified by western blot analysis and by indirect immunofluorescence with antibodies raised against Cdu1 (*Figure 1—figure supplement 2A and B*). Because *cdu2* resides directly downstream of the *cdu1* locus and encodes a second Ct DUB (Cdu2/*Chla*DUB2) (*Misaghi et al., 2006*), we first determined whether the disruption of *cdu1* impacted the expression of *cdu2*. We detected *cdu1* and *cdu2* transcripts in HeLa cells infected with Ct L2 but not for the juncture between *cdu1* and *cdu2* (*Figure 1—figure supplement 2C*). In cells infected with Ct *cdu1*::GII we only detected *cdu2* transcripts (*Figure 1—figure supplement 2C*) confirming that *cdu1* and *cdu2* are not co-expressed as part of an operon in accordance with previous observations (*Albrecht et al., 2010*).

In transfected HeLa cells, Cdu1's DUB activity has been linked to fragmentation of the Golgi apparatus (*Pruneda et al., 2018*). We therefore hypothesized that Cdu1s' DUB activity in infected cells promoted Golgi redistribution around inclusions and that we could identify potential targets by comparing the protein ubiquitination profile of cells infected with WT or *cdu1*::GII strains by quantitative mass spectrometry (MS). HeLa cell were mock infected or infected with either WT L2 or *cdu1*::GII strains. At 24 hpi, poly-ubiquitinated proteins were enriched from lysed cells using Tandem Ubiquitin Binding Entities (TUBEs; LifeSensors). TUBEs consist of concatenated Ub binding associated domains (UBAs) that bind to polyUb-modified proteins with nanomolar affinities. Poly-ubiquitinated proteins of both human and Ct origin were enriched and identified by quantitative LC-MS/MS analysis.

Over 2000 non-ubiquitinated proteins co-precipitated with TUBE1-bound proteins across all three conditions (mock, L2, and *cdu1*::GII infected HeLa cells) and three biological replicates (*Supplementary file 1*). Among these, 47 human proteins were significantly enriched in mock infected HeLa cells and 50 human proteins were significantly enriched during Ct infection (L2 and *cdu1*::GII) (*Supplementary file 3*, *Figure 1—figure supplement 3*). Pathway enrichment analysis revealed that proteins involved in RNA metabolism were overrepresented among co-precipitating proteins from mock infected cells (*Supplementary file 4*, *Figure 1—figure supplement 4*) while no biological pathways or processes were overrepresented in proteins enriched from infected cells (*Figure 1—figure supplement 4*). We also identified eight TUBE1 co-precipitating Ct proteins in HeLa cells infected with L2 and *cdu1*::GII (*Supplementary file 5*, *Figure 1—figure supplement 3*).

TUBE1 affinity capture lead to the identification of 43 ubiquitinated proteins (35 human proteins and 8 Ct proteins across all 3 conditions and replicates) based on the presence of peptides containing a di-glycine remnant motif (*Peng et al., 2003*; *Supplementary files 6-8*). The lack of widespread poly-ubiquitination of either human or Ct proteins in response to Ct infection (*Figure 1*) was surprising

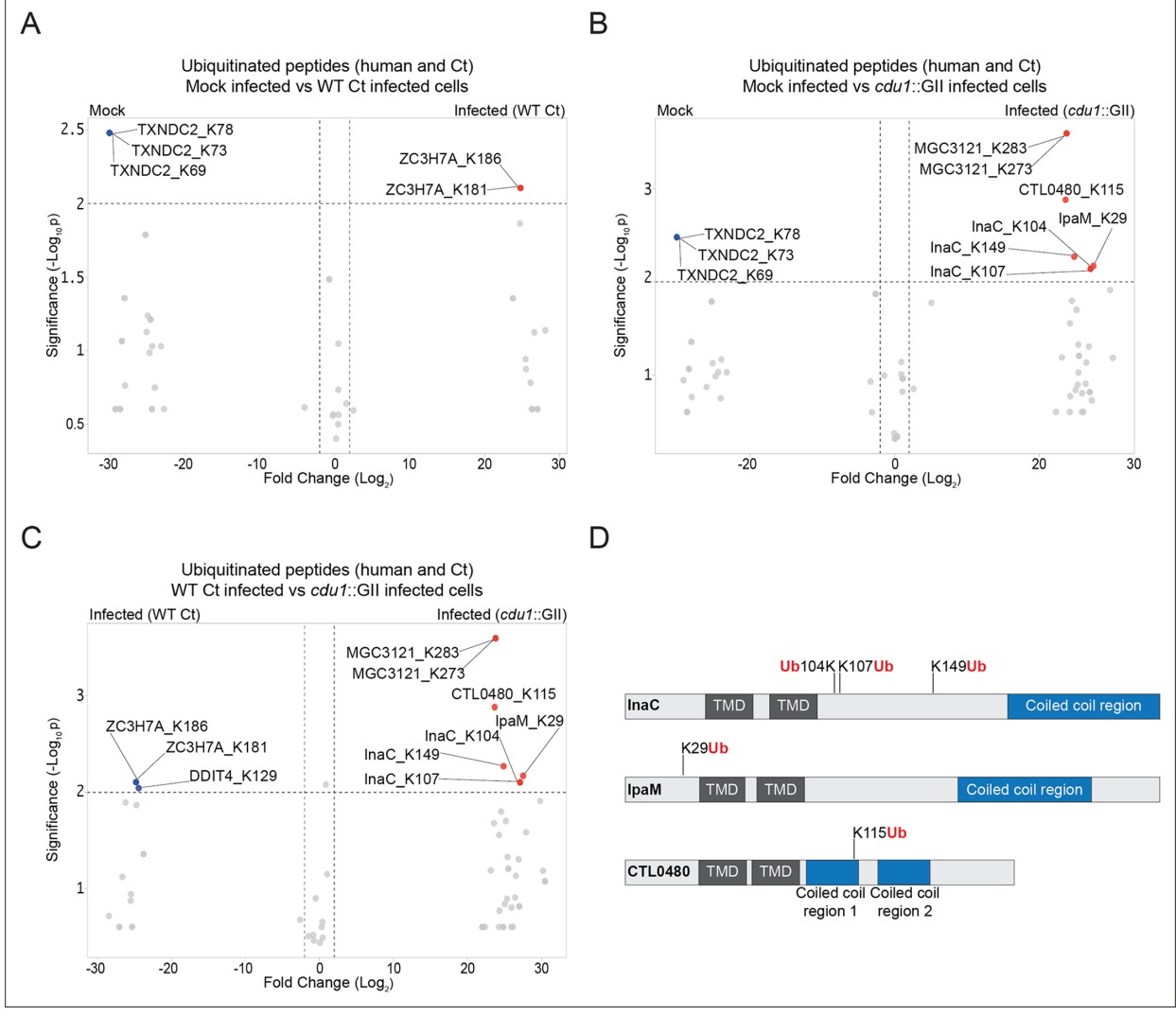

**Figure 1.** The *C. trachomatis* inclusion membrane proteins InaC, IpaM, and CTL0480 are ubiquitinated in the absence of Cdu1. (**A–C**) Volcano plots (pairwise comparisons) of the relative abundance of human and Ct ubiquitinated peptides.(**A**) Mock infected HeLa cells versus HeLa cells infected with WT Ct (L2 434 Bu pBOMB) (24 hpi). (**B**) Mock infected HeLa cells versus HeLa cells infected with a *cdu1* null strain (*cdu1*::GII pBOMB) (24 hpi) (**C**) HeLa cells infected with WT Ct (24 hpi) versus HeLa cells infected with a *cdu1* null strain (24 hpi). Significance values were interpolated from 3 independent biological replicates. Ubiquitinated proteins were enriched with magnetic TUBE1 beads (binds to polyubiquitinated proteins) and peptides identified by quantitative LC MS/MS analysis. Three Ct inclusion membrane proteins, InaC, IpaM, and CTL0480 were differentially ubiquitinated in the absence of Cdu1. (**D**) InaC was ubiquitinated at K104, K107, and K149, IpaM at K290, and CTL0480 at K115 in the absence of Cdu1. TMD: Transmembrane domain. Numbering corresponds to amino acids in the protein sequence of each respective inclusion membrane protein.

The online version of this article includes the following source data and figure supplement(s) for figure 1:

**Source data 1.** Excel file containing MaxQuant intensity values used to generate volcano plots in *Figure 1A–C*.

**Figure supplement 1.** TargeTron mediated disruption of the L2 *cdu1* ORF.

**Figure supplement 2.** Generation of a *cdu1* null strain in *C. trachomatis* (L2).

**Figure supplement 2—source data 1.** Original files used for western blot analysis shown in *Figure 1—figure supplement 2A* (anti-Cdu1, anti-RpoB, and anti-α-tubulin).

**Figure supplement 2—source data 2.** Files containing original scans of western blot analysis shown in *Figure 1—figure supplement 2A* (anti-Cdu1, anti-RpoB, and anti-α-Tubulin) with sample labels and highlighted bands.

**Figure supplement 3.** Proteins co-precipitating (non ubiquitinated) with human and Ct proteins enriched by TUBE1 pulldowns.

**Figure supplement 3—source data 1.** Excel file containing MaxQuant intensity values used to generate volcano plots in panels A and B.

**Figure supplement 4.** Pathway enrichment analysis of human and Ct proteins that co-precipitate (non ubiquitinated) with TUBE1-bound proteins.

*Figure 1 continued on next page*

*Figure 1 continued*

**Figure supplement 4—source data 1.** Excel file with Metascape compiled GO clusters and associated p-values for GO analysis of human co-precipitating proteins in mock-infected Hela cells (panel A).

**Figure supplement 4—source data 2.** Excel file with Metascape compiled GO clusters and associated p-values for GO analysis of human co-precipitating proteins in *cdu1* null infected HeLa cells (panel B).

given that wholesale changes in protein ubiquitination has been reported during infection of HeLa cells by intracellular pathogens like *S.* Typhimurium (*Fiskin et al., 2016*). It is also possible that had we conducted our analysis at different time points post-infection (hpi), we might have identified additional Cdu1 targets, such as Mcl1 and IκBα (*Le Negrate et al., 2008*; *Fischer et al., 2017*) which were not identified in our analysis. However, given that we observed Cdu1 at the inclusion membrane as early as 1 hpi (data not shown), we opted to focus on an earlier stage of the infection cycle. Only two human ubiquitinated proteins (ZC3H7A and DDIT4) were found to be significantly enriched in response to WT L2 infection at 24 hpi (*Figure 1A and C*, *Supplementary file 8*) while only one human protein (MGC3121) was preferentially ubiquitinated in HeLa cells infected with the *cdu1*::GII mutant strain (*Figure 1B and C*, *Supplementary file 8*). In contrast three Ct proteins, InaC (K104, K107, and K149), IpaM (K29), and CTL0480 (K115) were ubiquitinated at Lys residues in the absence of Cdu1 (*Figure 1B, C and D*, *Supplementary file 8*).

InaC, IpaM, and CTL0480 are Ct effector proteins that localize to the inclusion membrane (*Chen et al., 2006*; *Alzhanov et al., 2009*; *Lutter et al., 2013*; *Kokes et al., 2015*). These Type 3 secretion substrates belong to a family of over 36 inclusion membrane proteins (Incs) that contain a signature bi-lobal hydrophobic transmembrane domain (*Bannantine et al., 2000*; *Rockey et al., 2002*; *Li et al., 2008*; *Dehoux et al., 2011*; *Lutter et al., 2012*). Incs provide many functions important for Ct intracellular replication ranging from providing structural integrity to the inclusion membrane, to regulating membrane trafficking, to mediating interactions with host organelles and cytoskeletal structures (reviewed in *Bugalhão and Mota, 2019*). InaC facilitates the activation of the small GTPase RhoA, a crucial step for the assembly of actin scaffolds around the inclusion (*Haines et al., 2021*; *Kumar and Valdivia, 2008*). Additionally, InaC plays a pivotal role in the activation of Arf GTPases. This activation subsequently induces PTMs of microtubules in close proximity to the inclusion membrane, which are essential for Ct to initiate the repositioning of Golgi ministacks around the inclusion (*Wesolowski et al., 2017*). IpaM exhibits localization to discrete patches in the inclusion termed microdomains (*Alzhanov et al., 2009*; *Dumoux et al., 2015*). Upon ectopic expression, IpaM induces alterations in microtubule organization (*Dumoux et al., 2015*). CTL0480, facilitates the recruitment of MYPT1 (myosin phosphatase target subunit 1) to the inclusion membrane. Recrutment of MYPT1 is required for the efficient exit of Ct from host cells (*Lutter et al., 2013*; *Shaw et al., 2018*). Because Cdu1 also localizes to the inclusion membrane (*Fischer et al., 2017*; *Wang et al., 2018*; *Pruneda et al., 2018*; *Kunz et al., 2019*) we postulated that Cdu1 directly protects InaC, IpaM, and CTL0480 from ubiquitination.

## Cdu1 associates with InaC, IpaM, and CTL0480

We first determined if Cdu1 co-localized with InaC, IpaM, and CTL0480 at the inclusion membrane. HeLa cells were infected for 24 hr with WT L2 or L2 expressing CTL0480-Flag from its endogenous promoter and immunostained with antibodies against Cdu1, InaC, IpaM, or the Flag epitope. Both InaC and CTL0480-Flag localized throughout the inclusion membrane while IpaM was restricted to discrete microdomains as previously reported (*Alzhanov et al., 2009*; *Dumoux et al., 2015*; *Figure 2A*). Cdu1 co-localized with InaC and CTL0480-Flag and with IpaM at microdomains (*Figure 2A and B*). All four antibodies specifically recognized their corresponding antigens since immunostaining for Cdu1, InaC, IpaM or the Flag epitope was not observed in Ct strains lacking Cdu1 (*cdu1*::GII), InaC (M407, *Kokes et al., 2015*) IpaM (*ipaM*::GII, *Meier et al., 2023*), or a strain that does not express CTL0480-Flag (*Figure 2A*).

We next determined if Cdu1 can interact with InaC, IpaM, and CTL0480 by co-transfecting HEK 293 cells with vectors expressing either full length Cdu1-GFP or truncated versions of Cdu1-GFP lacking transmembrane or catalytic domains (*Figure 2C*), and vectors expressing Flag-InaC, V5-IpaM, and V5-CTL0480. Transfected cells were lysed and Cdu1-GFP was immunoprecipitated with antibodies against GFP. Western blot analysis of the immunoprecipitates showed that Flag-InaC, V5-IpaM, and

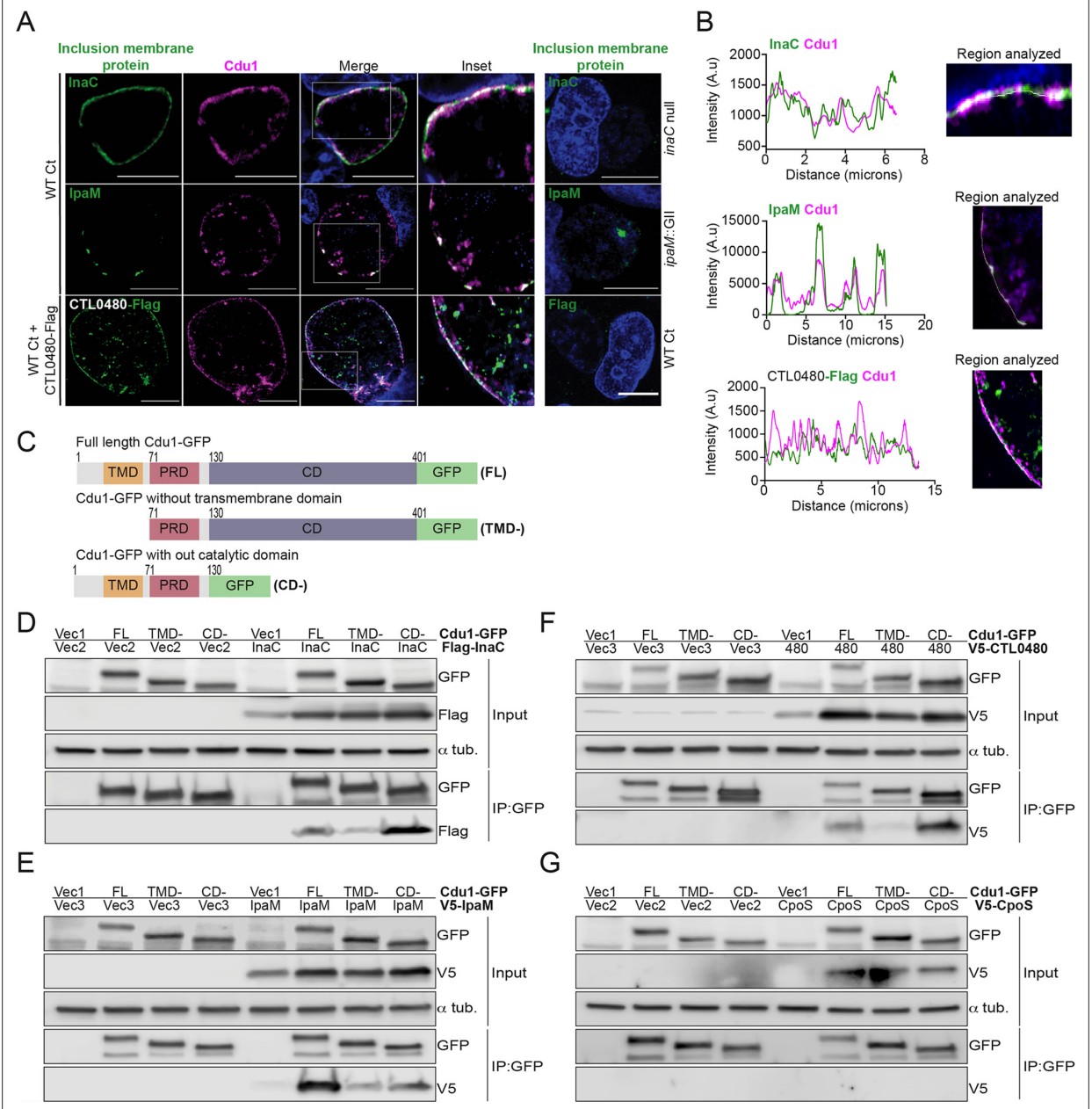

**Figure 2.** Cdu1 associates with InaC, IpaM, and CTL0480. (**A**) Co-localization of Cdu1(magenta) with endogenous InaC (green), IpaM (green), and ectopically expressed CTL0480-Flag (green) at the Ct (L2) inclusion membrane of HeLa cells infected for 24 hr. HeLa cells infected with an *inaC* null strain (M407), an *ipaM* null strain (*ipaM*::GII), and WT Ct (L2 434 Bu pBOMB) were used as controls for antibody specificity. DNA stained with Hoechst is shown in blue. Scale bar: 10 μm. Images are representative of multiple images captured across three independent replicates. (**B**) Line scan profiles of fluorescent signal intensities displayed in (**A**) showing co-localization of fluorescence intensities for endogenous Cdu1 with endogenous InaC and IpaM, and with CTL0480-Flag along the L2 inclusion membrane. (**C**) Schematic of Cdu1-GFP(C) (L2) fusion (Cdu1-GFP) and Cdu1-GFP variants used in co-transfections of HEK 293 cells. GFP: Green fluorescent protein. TMD: Transmembrane domain. PRD: Proline rich domain. CD: Catalytic domain. FL: Full length. TMD-: Cdu1-GFP variant lacking TMD domain. CD-: Cdu1-GFP variant lacking CD domain. (**D–G**) Western blot analysis of GFP immunoprecipitates from HEK 293 cells co-transfected with mammalian plasmids expressing: Cdu1-GFP variants and (**D**) truncated 3XFlag(N)-InaC (D/UW-3/CX CT813, amino acids 96–264), (**E**) V5(N)-IpaM (L2, full length), (**F**) V5(N)-CTL0480 (L2, full length), and (**G**) V5(N)-CpoS (L2, full length). Vec1: Empty pOPINN-GFP vector. Vec2: Empty pDEST53 vector. Vec3: Empty pcDNA3.1/nV5-DEST vector. Western blot images are representative from two independent experiments.

The online version of this article includes the following source data for figure 2:

**Source data 1.** Excel files with pixel intensity values for generating line scan profiles in *Figure 2B*.

*Figure 2 continued on next page*

*Figure 2 continued*

**Source data 2.** Original western blot scans used to generate *Figure 2D* (anti-GFP [input], anti-Flag [input], anti-α-tubulin [input], anti-GFP [IP:GFP], and anti-Flag [IP:GFP]).

**Source data 3.** Original western blot scans with labeled samples and highlighted bands for *Figure 2D*.

**Source data 4.** Original western blot scans used to generate *Figure 2E* (anti-GFP [input], anti-V5 [input], anti-α-tubulin [input], anti-GFP [IP:GFP], and anti-V5 [IP:GFP]).

**Source data 5.** Original western blot scans with labeled samples and highlighted bands for *Figure 2E*.

**Source data 6.** Original western blot scans used to generate *Figure 2F* (anti-GFP [input], anti-V5 [input], anti-α-tubulin [input], anti-GFP [IP:GFP], and anti-V5 [IP:GFP]).

**Source data 7.** Original western blot scans with labeled samples and highlighted bands for *Figure 2F*.

**Source data 8.** Original western blot scans used to generate *Figure 2G* (anti-GFP [input], anti-V5 [input], anti-α-tubulin [input], anti-GFP [IP:GFP], and anti-V5 [IP:GFP]).

**Source data 9.** Original western blot scans with labeled samples and highlighted bands for *Figure 2G*.

V5-CTL0480 co-precipitated with Cdu1-GFP (*Figure 2D–F*). Moreover, the transmembrane domain of Cdu1 was necessary for Cdu1-GFP to interact with all three Incs (*Figure 2D–F*). The interaction between Cdu1-GFP and the three tagged Incs was specific since we did not detect interactions between Cdu1-GFP and a V5-tagged version of the inclusion membrane protein CpoS (*Sixt et al., 2017*; *Figure 2G*). We expected these interactions to be transient at the inclusion membrane as the engagement of Cdu1 with its target(s) should mimic that of most enzymes with their substrates. Therefore, while we could capture these complexes in co-immunoprecipitations in the context of overexpression, this assay was not sensitive enough to reliably document the formation of complexes among low abundance endogenous Ct proteins. Nevertheless, our findings from transfection experiments lead us to conclude that Cdu1 can selectively interact with all three Incs even in the absence of infection, and these interactions are facilitated by the transmembrane domain of Cdu1.

## Cdu1 protects InaC, IpaM, and CTL0480 proteins from degradation during infection

We next assessed if Cdu1 was required to stabilize endogenous InaC, IpaM, and CTL0480 in infected cells. HeLa cells were infected with either WT L2 or *cdu1*::GII strains and at various time points in the Ct infectious cycle crude cell lysates were analyzed by western blot to assess the relative abundance of Inc proteins. At 36 and 48 hpi, the levels of InaC protein were found to be undetectable in cells infected with *cdu1*::GII in comparison to cells infected with the WT L2 strain (*Figure 3A and B*). Conversely, IpaM protein levels exhibited a decrease at 36 and 48 hpi in cells infected with *cdu1*::GII (*Figure 3C and D*). We were not able to detect CTL0480 levels by western blot but were successful in following CTL0480 expression by indirect immunofluorescence (*Figure 3E*). The relative abundance of CTL0480 at inclusion membranes was not affected in *cdu1*::GII inclusions at 24 hpi. However, at 36 hpi, a subpopulation of cells lost CTL0480 immunoreactivity and by 48 hpi, inclusion membranes of the *cdu1*::GII strain were devoid of CTL0480 while CTL0480 was prominently detected at the inclusion membranes of WT L2 (*Figure 3E* and *Figure 3—figure supplement 1*). As controls for the specificity of antibodies used for western blots and for indirect immunofluorescence, we included cells infected with Ct lacking *ipaM* (*ipaM*::GII, *Meier et al., 2023*), with an *inaC* nonsense mutant (M407, *Kokes et al., 2015*), and with Ct lacking *ctl0480* (*ctl0480*::GII, *Shaw et al., 2018*). Overall, our results indicate that steady state protein levels of InaC and IpaM and CTL0480 localization at the inclusion membrane, especially at late stages of infection, are dependent on Cdu1, and that Cdu1 acts at different stages in the infection cycle.

## The acetylase activity of Cdu1 is required for Cdu1 to protect itself, InaC, and IpaM from polyubiquitination and proteasomal degradation

The crystal structures of Cdu1 bound to Ub or Coenzyme A indicated that the adenosine and phosphate groups of Coenzyme A make contact with a helix in variable region 3 (VR-3) of Cdu1 while the Ile36-patch of Ub binds to the opposite face of the same helix (*Pruneda et al., 2018*). Although Cdu1 catalyzes both of its DUB and acetylase (Act) activities with the same active site (*Pruneda et al.,*

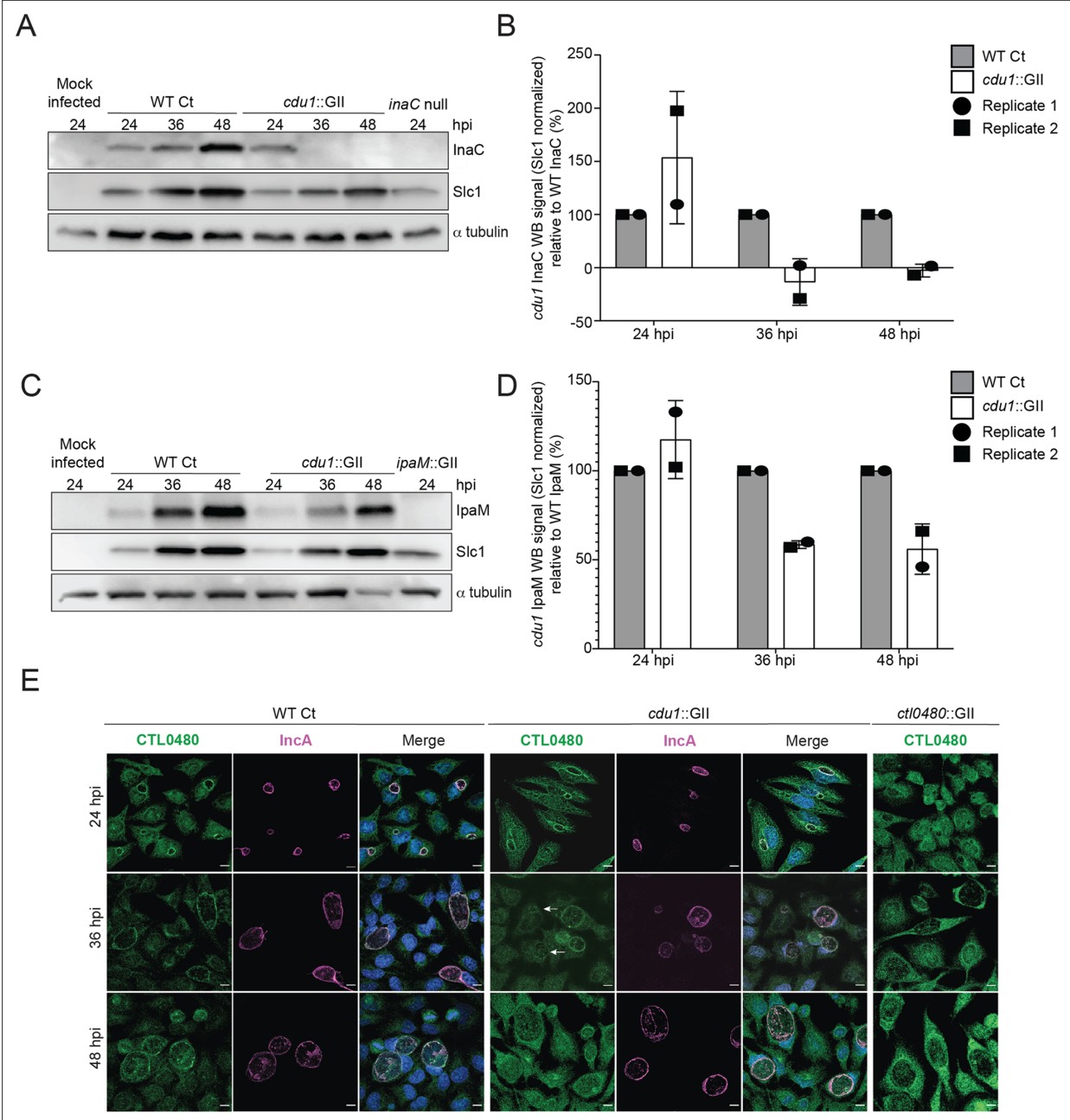

**Figure 3.** Cdu1 stabilizes InaC, IpaM, and CTL0480. (**A**) Western blot analysis of endogenous InaC in HeLa cells infected for 24, 36, and 48 hr with Wt Ct (L2 pBOMB), *cdu1* null (*cdu1*::GII pBOMB), and *inaC* null (M407) strains. Ct Slc1 and human alpha tubulin were used to determine Ct burdens and equal loading of protein extracts respectively. Western blot images are representative of 2 independent experiments. (**B**) Quantification of InaC abundance (InaC western blot signal from (**A**)) normalized to Slc1 western blot signal (from panel A) in HeLa cells infected with a *cdu1* null strain, relative to normalized InaC abundance in HeLa cells infected with Wt Ct. Error bars depict standard deviation. (**C**) Western blot analysis of endogenous IpaM in HeLa cells infected for 24, 36, and 48 hr with Wt Ct, *cdu1* null, and *ipam* null (*ipam*::GII) strains. Western blot images are representative of two independent experiments. (**D**) Quantification of normalized IpaM abundance from (**C**) in HeLa cells infected with a *cdu1* null strain, relative to normalized IpaM abundance in HeLa cells infected with Wt Ct. Error bars depict standard deviation. (**E**) Localization of CTL0480 during Ct infection of HeLa cells at 24, 36, and 48 hpi. CTL0480 signal (green) co-localizes with the inclusion membrane protein IncA (magenta) at the Ct inclusion membrane. Arrowheads highlight *cdu1* null inclusions lacking CTL0480 at 36 hpi. DNA stained with Hoechst is shown in blue. Scale bar: 10 μm. Quantification of CTL0480 localization at inclusion membranes can be found in *Figure 3—figure supplement 1*.

The online version of this article includes the following source data and figure supplement(s) for figure 3:

**Source data 1.** Original western blot scans used to generate *Figure 3A* (anti-InaC, anti-Slc1, and anti-α-tubulin).

*Figure 3 continued on next page*

*Figure 3 continued*

**Source data 2.** Original western blot scans with labeled samples and highlighted bands for *Figure 3A*.

**Source data 3.** Excel file containing densitometry data for *Figure 3B*.

**Source data 4.** Original western blot scans used to generate *Figure 3C* (anti-IpaM, anti-Slc1, and anti-α-tubulin).

**Source data 5.** Original western blot scans with labeled samples and highlighted bands for *Figure 3C*.

**Source data 6.** Excel file containing densitometry data for *Figure 3D*.

**Figure supplement 1.** Quantification of CTL0480 at Ct inclusion membranes.

**Figure supplement 1—source data 1.** Excel files with numerical data for quantifying CTL0480 localization to WT and *cdu1*::GII inclusions.

*2018*) the two activities of Cdu1 can be uncoupled by the amino acid substitution K268E in VR-3 which disrupts Coenzyme A binding required for Act activity and by the amino acid substitution I225A in the Ub-binding region of VR-3 required for DUB activity (*Pruneda et al., 2018*). These substitutions allowed us to test which of Cdu1's enzymatic functions are required for the observed effects on protein stability. We generated Ct shuttle plasmids, expressing WT Cdu1, a catalytically inactive variant of Cdu1 lacking both DUB and Act activities (Cdu1$^{C345A}$; *Pruneda et al., 2018*), a Cdu1 DUB-deficient variant (Cdu1$^{I225A}$), and a Cdu1 Act-deficient variant (Cdu1$^{K268E}$). All Cdu1 constructs were expressed from the *cdu1* endogenous promoter as 3 X Flag epitope-tagged proteins.

Plasmids expressing each Cdu1 variant were transformed into the *cdu*1::GII mutant and the resulting strains used to infect HeLa cells for 36 and 48 hr. The levels of endogenous InaC and IpaM in cell extracts of infected cells were monitored by western blot analysis. At 36 hpi, InaC protein levels drastically decreased in cells infected with *cdu1* null strains transformed with empty vector or expressing the catalytic inactive variant of Cdu1 (Cdu1$^{C345A}$-Flag; *Figure 4A*). Likewise, IpaM protein levels diminished at 48 hpi during infection with the same strains (*Figure 4B*). Both InaC and IpaM protein levels were restored to wild type levels in *cdu1* null strains complemented with wild type Cdu1-Flag (*Figure 4A and B*). Unexpectedly, cells infected with a *cdu1* null strain ectopically expressing the DUB-deficient Cdu1 variant (Cdu1$^{I225A}$-Flag) displayed wild type levels of InaC and IpaM while the Act-deficient variant (Cdu1$^{K268E}$-Flag) did not (*Figure 4A and B*). These results suggest that the acetylase activity of Cdu1 rather than its DUB activity is required for Cdu1's ability to stabilize InaC and IpaM proteins.

When we monitored the stability of each Flag-tagged Cdu1 variant, we observed that the catalytically inactive variant of Cdu1 (C345A-Flag) was destabilized (*Figure 4A and B*- Flag WB). We reasoned that Cdu1 also protects itself from being targeted for degradation in infected cells. As with InaC and IpaM, the acetylase but not the DUB activity of Cdu1 was required for Cdu1's stability (*Figure 4A and B*). Although the C345A (DUB-, Act-) and K268E (Act-) amino acid substitutions in Cdu1 do not destabilize Cdu1 expressed in *E. coli* (*Pruneda et al., 2018*), it was possible that these substitutions impacted the expression and/or folding of Cdu1 in *Chlamydia*. To determine if these Cdu1 mutants were inherently unstable, we expressed Cdu1$^{C345A}$-Flag and Cdu1$^{K268E}$-Flag in WT L2 in the presence of endogenous Cdu1. We found that each Flag-tagged variant was stabilized (*Figure 4C*), indicating that endogenous Cdu1 protected the catalytically-deficient Cdu1 variants in trans. In addition, western blot analysis of immunoprecipitated Cdu1-Flag and Cdu1$^{C345A}$-Flag expressed in a *cdu1*::GII mutant showed that while WT Cdu1-Flag was not modified by Lys48-linked poly-ubiquitination, Cdu1$^{C345A}$-Flag was robustly modified by Lys48-linked polyUb in the presence of the proteasome inhibitor MG132 (*Figure 4D*). Moreover, endogenous Cdu1 protected Cdu1$^{C345A}$-Flag from Lys48-linked polyUb when Cdu1$^{C345A}$-Flag was expressed in a wild type L2 background (*Figure 4D*). These results indicate that the loss of Cdu1 activity likely leads to its Lys48-linked poly-ubiquitination and subsequent proteasome-dependent degradation.

Because Cdu1 autoacetylates itself and its Act activity is directed towards lysines (*Pruneda et al., 2018*) we postulated that Cdu1 may stabilize proteins from degradation by acetylating lysine residues that are potential targets of ubiquitination. We tested this hypothesis by assessing whether Cdu1, InaC, CTL0480, and IpaM are acetylated at lysines during infection. Fractions enriched for inclusion membranes were isolated by sub-cellular fractionation from HeLa cells infected with wild-type L2 (24 hpi), and proteins acetylated at lysines were immunoprecipitated. Western blot analysis of acetyl-lysine immunoprecipitates indicated that InaC and IpaM, but not the Inc protein IncA, were acetylated (*Figure 4E*). Western blot analysis of anti-acetyl-lysine immunoprecipitates of inclusion

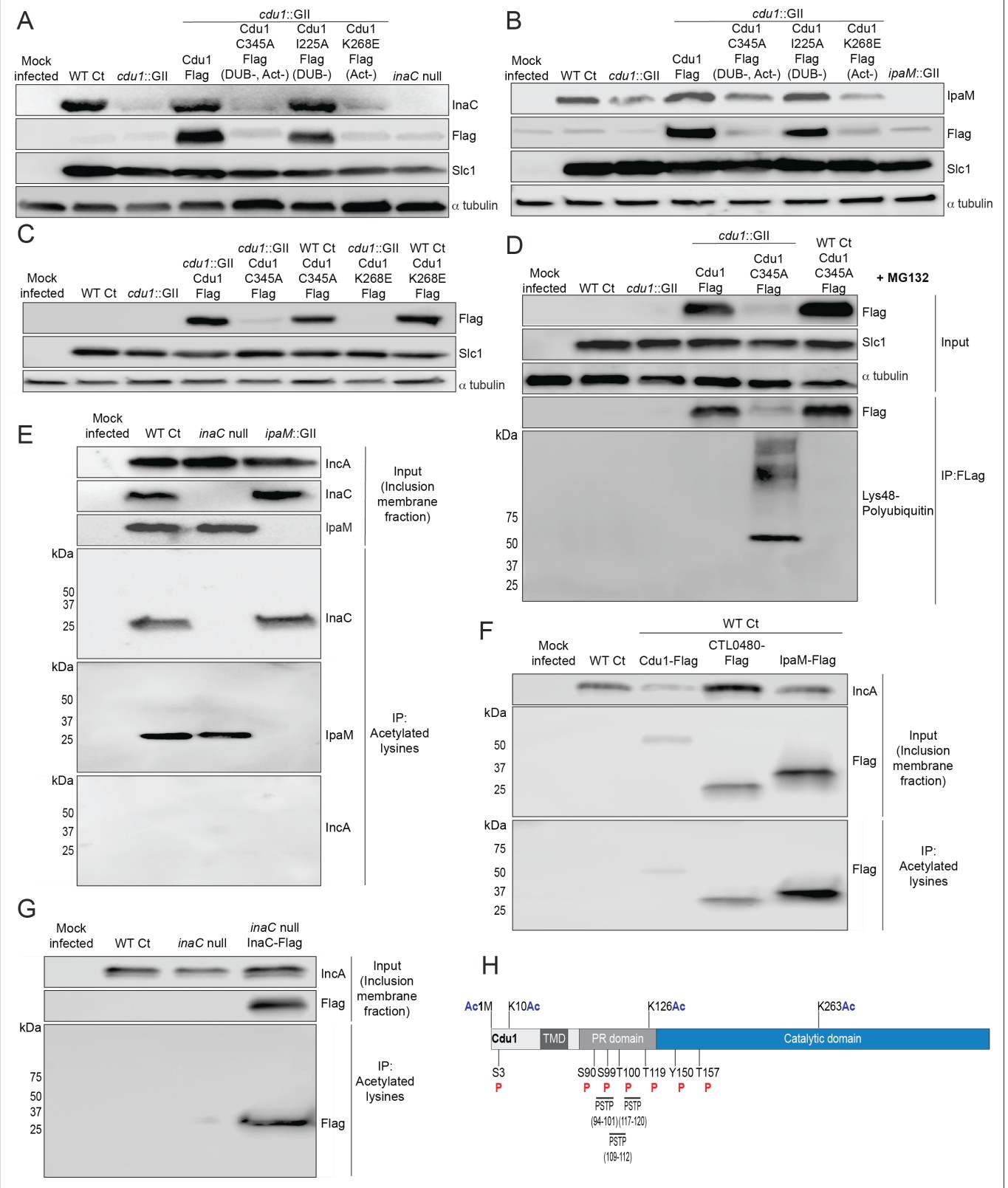

**Figure 4.** The acetylase activity of Cdu1 is required to stabilize Cdu1, InaC, and IpaM. (**A**) Western blot analysis of endogenous InaC and Cdu1-Flag catalytic variants expressed from a plasmid (pBOMB). HeLa cells were infected for 36 hr with WT Ct (L2 434 Bu pBOMB), a *cdu1* null strain (*cdu1*::GII pBOMB), and *cdu1* null strains expressing wild type Cdu1-Flag and the Cdu1 variants C345A-Flag (catalytic inactive), I225A-Flag (DUB deficient), and K268E-Flag (Act deficient). Cdu1-Flag variants were expressed from a pBOMB shuttle plasmid. Protein lysates from HeLa cells infected with an

*Figure 4 continued*

*inaC* null (M407) strain were used to control for the specificity of anti-InaC antibodies. Western blot images are representative of two independent experiments. (**B**) Western blot analysis of endogenous IpaM and Cdu1-FLAG variants in crude extracts of HeLa cells infected for 48 hr with the same strains as describe in (**A**). Infection of HeLa cells with *ipaM*::GII was used to test for the specificity of the anti-IpaM antibody. Western blot images are representative of two independent experiments. (**C**) Western blot analysis of Cdu1$^{C345A}$-Flag (catalytic inactive) and Cdu1$^{K268E}$-Flag (Act deficient) expressed in a *cdu1* null strain or WT Ct (L2 434 Bu) background after infection of HeLa cells for 24 hr. Both Cdu1 variants are stabilized by Cdu1 expressed in WT Ct. (**D**) Western blot analysis of Cdu1-Flag and Cdu1$^{C345A}$-Flag expressed in a *cdu1* null strain and Cdu1$^{C345A}$-Flag expressed in WT Ct (L2 434 Bu) following immunoprecipitation (anti-Flag) from HeLa cell extracts after infection for 24 hr and treatment with MG132 (25 μM, 5 hr). Western blot image is a representative blot from at least three independent experiments. (**E**) Western blot analysis of endogenous InaC and IpaM following immunoprecipitation of inclusion membrane enriched subcellular fractions (24hpi) with anti-acetylated lysine antibodies. Western blot image is representative of two independent experiments. (**F**) Western blot analysis (Flag WB) of acetylated lysine immunoprecipitates generated from inclusion membrane enriched subcellular fractions (40 hpi) derived from HeLa cells infected with WT Ct strains expressing Cdu1-Flag, CTL0480-Flag, or IpaM-Flag. Western blot image is representative of two independent experiments. (**G**) WB of acetylated lysine immunoprecipitates of inclusion membrane enriched fractions (24 hpi) of HeLa cells infected with WT Ct and with an *inaC* null strain (M407) expressing InaC-Flag. Western blot image is representative of two independent experiments. (**H**) The initiator methionine, Lys10, Lys126, and Lys263 of Cdu1 are acetylated by 24 hpi. One tyrosine (Y) residue and multiple serine (S) and threonine (T) residues in Cdu1 are also phosphorylated during Ct infection of HeLa cells (24 hpi). Three PX(S/T)P MAPK phosphorylation consensus sequence motifs were identified in the proline rich domain of Cdu1. Modified residues were identified by quantitative LC MS/MS analysis of immunoprecipitated Cdu1-Flag across three independent biological replicates. TMD: Transmembrane domain. PR: Proline rich. PSTP: PX(S/T)P motifs. Ac: Acetylation. P: Phosphorylation. Numbering corresponds to amino acids in Cdu1 protein sequence.

The online version of this article includes the following source data for figure 4:

**Source data 1.** Original western blot scans used to generate *Figure 4A* (anti-InaC, anti-Flag, anti-Slc1, and anti-α-tubulin).

**Source data 2.** Original western blot scans with labeled samples and highlighted bands for *Figure 4A*.

**Source data 3.** Original western blot scans used to generate *Figure 4B* (anti-IpaM, anti-Flag, anti-Slc1, and anti-α-tubulin).

**Source data 4.** Original western blot scans with labeled samples and highlighted bands for *Figure 4B*.

**Source data 5.** Original western blot scans used to generate *Figure 4C* (anti-Flag, anti-Slc1, and anti-α-tubulin).

**Source data 6.** Original western blot scans with labeled samples and highlighted bands for *Figure 4C*.

**Source data 7.** Original western blot scans used to generate *Figure 4D* (anti-Flag [input], anti-Slc1 [input], anti-α-tubulin [input], anti-Flag [IP:Flag], and anti-lys48 pUb [IP:Flag]).

**Source data 8.** Original western blot scans with labeled samples and highlighted bands for *Figure 4D*.

**Source data 9.** Original western blot scans used to generate *Figure 4E* (anti-IncA [input], anti-InaC [input], anti-IpaM [input], anti-InaC [IP:Acetylated lysines], anti-IpaM [IP:Acetylated lysines], and anti-IncA [IP:Acetylated lysines]).

**Source data 10.** Original western blot scans with labeled samples and highlighted bands for *Figure 4E*.

**Source data 11.** Original western blot scans used to generate *Figure 4F* (anti-IncA [input], anti-Flag [input], and anti-Flag [IP:Acetylated lysines]).

**Source data 12.** Original western blot scans with labeled samples and highlighted bands for *Figure 4F*.

**Source data 13.** Original western blot scans used to generate *Figure 4G* (anti-IncA [input], anti-Flag [input], and anti-Flag [IP:Acetylated lysines]).

**Source data 14.** Original western blot scans with labeled samples and highlighted bands for *Figure 4G*.

---

membrane-enriched membrane fractions of HeLa cells infected with L2 expressing Cdu1-Flag (24 hpi), CTL0480-Flag (40 hpi), and IpaM-Flag (40 hpi) also showed that all three Flag tagged effectors were acetylated at lysines (*Figure 4F*). We also determined that Flag-tagged InaC expressed in an *inaC* null (M407) background was acetylated at lysines as determined by western blot analysis of anti-acetyl-lysine immunoprecipitates (*Figure 4G*). In addition, we identified acetylated forms of Cdu1 from mass spectrometric analysis of Flag immunoprecipitates derived from extracts of HeLa cells infected with L2 expressing Cdu1-Flag, (24 hpi) (*Figure 4H*).

## Cdu1's acetylase activity shields inclusions from ubiquitination but is not sufficient to protect against IFNγ mediated antimicrobial activity

We reasoned that the lysine acetylase activity of Cdu1 is a prominent mechanism by which Cdu1 protects client proteins (at 24 hpi), since loss of Cdu1 or expression of the acetylase-deficient variant of Cdu1 (Cdu1$^{K268E}$-Flag) leads to a marked increase in Ub immunostaining at or near the periphery of *cdu1*::GII inclusions (>80% of inclusions) compared to HeLa cells infected with *cdu1*::GII strains complemented with wild type or DUB-deficient (I225A-Flag) Cdu1 strains (*Figure 5A and B*, and *Figure 5—figure supplement 1A*). Given that Cdu1 appears to localize exclusively at inclusion membranes, we predicted that its activity would be spatially restricted to the inclusion periphery. We tested this

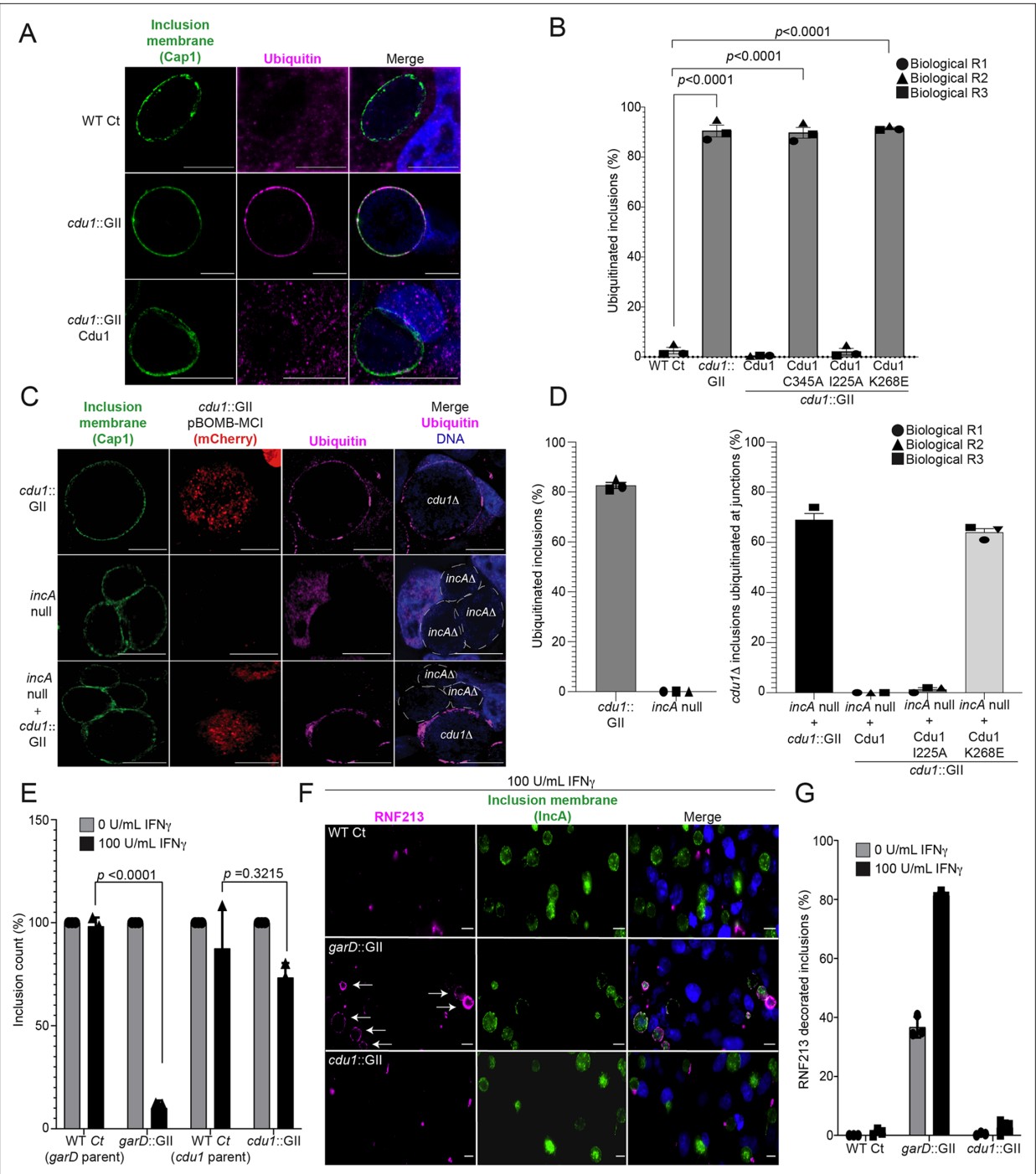

**Figure 5.** The DUB activity of Cdu1 is not required for blocking the ubiquitination of inclusion membranes and Cdu1 is not required for protection against IFNγ mediated cellular immunity. (**A**) Representative images of Ct inclusions decorated with ubiquitin during infection of HeLa cells with a *cdu1*::GII strain for 24 hr. Representative images of infected HeLa cells used for quantification of ubiquitin decorated inclusions in (**B**) are shown in *Figure 5—figure supplement 1A*. Antisera against the membrane protein Cap1 (green) was used to mark inclusion membranes. DNA stained with Hoechst is shown in blue. Scale bar: 10 μm. (**B**) Quantification of ubiquitinated inclusions as shown in (**A**). The Ub fluorescent signal was used to determine the number of infected cells with Ub decorated inclusions. The total number of ubiquitinated inclusions was divided by the total number of inclusions analyzed (defined by Cap1 staining). 87%, 86%, and 91% of inclusions were decorated with Ub in HeLa cells infected with a *cdu1* null strain and *cdu1* null strains expressing Cdu1[C345A] (DUB-, Act-), and Cdu1[K268E] (Act-) variants respectively. Representative images (panel (**A**) and *Figure 5—figure supplement 1A*) and quantification of ubiquitinated inclusions were obtained from inclusions imaged in 10 fields across 3 independent biological replicates for each strain. p values were determined by a student paired t-test. For cells infected with the strains WT Ct, *cdu1*::GII, and *cdu1*::GII strains expressing WT Cdu1 and the variants C345A, I225A and K268E, n (across 3 replicates) = 262, 1505, 652, 1229, 389, and 1413 respectively.

*Figure 5 continued on next page*

*Figure 5 continued*

Error bars depict standard deviation.**(C)** Representative images of HeLa cells co-infected with *cdu1*::GII (*cdu1*Δ) and *incA* null (*incA*Δ, M923) strains at 24 hpi. IncA-deficient inclusions do not fuse with other inclusions. In co-infected cells, Cdu1 present on the inclusion membranes of *incA*Δ strains did not block ubiquitination events at or near the inclusion membranes of neighboring *cdu1*Δ strains (mCherry signal from pBOMB4-MCI plasmid). Representative images of HeLa cells infected with strains quantified in **(D)** are shown in *Figure 5—figure supplement 1B*. DNA stained with Hoechst is shown in blue. Scale bar: 10 µm. **(D)** Quantification of *cdu1*Δ inclusions as shown in **(C)** in which ubiquitination events are observed in regions of *cdu1*Δ inclusion membranes that are in direct apposition to *incA*Δ inclusion membranes (junctions). The total number of *cdu1*Δ inclusions ubiquitinated at inclusion junctions was divided by the total number of inclusions analyzed (Cap1 staining). 66% and 61% of *cdu1*Δ inclusions were decorated with Ub at junctions in HeLa cells co-infected with *incA*Δ and *cdu1*Δ strains or with *incA*Δ and a *cdu1*Δ strain ectopically expressing Cdu1$^{K268E}$ (Act-) respectively. Representative images (panel **(C)** and *Figure 5—figure supplement 1B*) and quantification of *cdu1*Δ inclusions ubiquitinated at junctions are derived from inclusions imaged in six fields across three independent biological replicates for each condition. p values were determined by a student paired t-test. For cells infected with the strains *cdu1*::GII and *incA* null, n (across 3 replicates) = 547 and 420 respectively. For cells co-infected with the *incA* null strain and *cdu1*::GII, or *cdu1*::GII strains expressing WT Cdu1 and the variants C345A, I225A and K268E, n (across 3 replicates) = 234, 188, 165, and 207 respectively. Error bars depict standard deviation. **(E)** Quantification of Ct inclusion production during infection of unprimed and IFNγ-primed (100 U/mL) A549 cells at 24 hours post infection. Inclusions were quantified by high-content imaging analysis. Plot reflects inclusion counts across nine fields of view and three independent biological replicates. Inclusion counts by each strain in unprimed A549 cells were set to 100%. Inclusion counts resulting from *cdu1*::GII and *garD*::GII strains were normalized to corresponding parental Ct inclusion (100%) counts in unprimed cells. p-values were calculated by two-way ANOVA analysis. For unprimed A549 cells infected with strains WT Ct (*garD*::GII parent), *garD*::GII, WT Ct (*cdu1*::GII parent), and *cdu1*::GII, n (across 3 replicates) = 20,212, 15,356, 10,931, and 9,535 respectively. For A549 cells pretreated with 100 U/mL IFNγ and infected with strains WT Ct (*garD*::GII parent), *garD*::GII, WT Ct (*cdu1*::GII parent), and c*du1*::GII, n (across 3 replicates) = 19,082, 1,170, 7,842, and 6,368 respectively. Error bars depict standard deviation. **(F)** Representative images of RNF213 localizing to inclusions of WT Ct, *garD*::GII, and *cdu1*::GII strains during infection of A549 cells primed with IFNγ (100 U/mL). **(G)** Quantification of RNF213 localizing to Ct inclusions during infection of unprimed and IFNγ-primed (100 U/mL) A549 cells at 24 hours post infection. Plot reflects inclusion counts across six fields and three independent biological replicates. For unprimed A549 cells infected with strains WT Ct, *garD*::GII, and *cdu1*::GII, n (across 3 replicates) = 830, 817, and 620 respectively. For A549 cells pretreated with 100 U/mL IFNγ and infected with strains WT Ct, *garD*::GII, and c*du1*::GII, n (across 3 replicates) = 816, 520, and 617 respectively. Error bars depict standard deviation.

The online version of this article includes the following source data and figure supplement(s) for figure 5:

**Source data 1.** Excel file with numerical data for quantifying ubiquitinated inclusions in HeLa cells infected with WT CT and *cdu1*::GII strains, and *cdu1*::GII strains expressing Cdu1 C345A, I225A, and K268E variants (*Figure 5B*).

**Source data 2.** Excel files with numerical data for quantifying the number of ubiquitinated inclusions in HeLa cells infected with *cdu1*::GII and *incA* null strains, and *cdu1*::GII strains expressing Cdu1 variants that are ubiquitinated at inclusion junctions in co-infected HeLa cells (*Figure 5D*).

**Source data 3.** Excel file containing numerical data for quantifying the number of inclusions in IFN-γ-primed A549 cells infected with WT CT, *garD*::GII, and *cdu1*::GII strains (*Figure 5E*).

**Source data 4.** Excel file containing numerical data for quantifying the number of RNF213-positive inclusions in IFN-γ-primed A549 cells infected with WT CT, *garD*::GII, and *cdu1*::GII strains (*Figure 5G*).

**Figure supplement 1.** The Acetylase activity of Cdu1 is the predominant activity of Cdu1 responsible for protecting Ct inclusions from ubiquitination.

premise by co-infecting HeLa cells with an *incA* null strain (M923 (IncA$^{R197*}$), *Kokes et al., 2015*) and the *cdu1*::GII strain. IncA mediates homotypic fusion of inclusion membranes and loss of IncA results in the accumulation of multiple unfused inclusions in cells infected at high MOIs (*Hackstadt et al., 1999*; *Suchland et al., 2000*; *Pannekoek et al., 2005*; *Figure 5C*). As expected, *incA* mutants which retain Cdu1 activity did not accumulate Ub at or near the periphery of inclusion membranes. In HeLa cells coinfected with both *cdu1*::GII and M923 (IncA$^{R197*}$), *cdu1*::GII Cdu1$^{C345A}$ and M923, or *cdu1*::GII Cdu1$^{K268E}$ and M923, only the *incA* null inclusions were protected from ubiquitination (*Figure 5C and D*, and *Figure 5—figure supplement 1B*). Based on these observations, we conclude that the acetylase activity of Cdu1 protects proteins in cis and that this activity is constrained to the membrane of the pathogenic vacuole consistent with previous reports (*Auer et al., 2020*).

Recently, the Ct inclusion membrane protein GarD was identified as a Ct effector that shields Ct from γ-interferon mediated ubiquitination by the IFNγ-inducible human ubiquitin E3 ligase RNF213 (*Walsh et al., 2022*). Because Cdu1 also protects the Ct inclusion from ubiquitination, we tested if Cdu1 also plays a role in protecting Ct from IFNγ-induced cell immunity. A549 cells were pretreated with IFNγ (100 U/mL) and infected with WT Ct, *cdu1*::GII, or *garD*::GII strains. Infections with a *garD*::GII strain led to an approximate 90% decrease in the number of inclusions formed relative to infections with its parental WT Ct strain (*Figure 5E*) while infections with a *cdu1*::GII or its parental WT Ct strain showed a modest reduction in inclusion formation (approximately 26% and 12%, respectively) (*Figure 5E*).

These results suggest that Cdu1 likely does not play a role in protecting Ct from IFNγ-mediated cellular immunity.

We also tested whether RNF213 localizes to inclusions that lack Cdu1, as observed in IFNγ-primed A549 cells infected with *garD*::GII strains (*Walsh et al., 2022*). RNF213 did not localize to Ct inclusions when cells were infected with either WT Ct or *cdu1*::GII strains, regardless of whether or not the A549 cells were treated with IFNγ (*Figure 5F and G*). In contrast, RNF213 localized to approximately 37% of inclusions in cells infected with *garD*::GII mutants of unprimed A549 cells, and 81% in IFNγ-treated cells (*Figure 5F and G*). Based on these results, we conclude that Cdu1 does not play a role in protecting Ct from IFNγ-induced antimicrobial activity.

## Cdu1 is required for F-actin assembly and Golgi ministack repositioning around the Ct inclusion, and for MYPT1 recruitment to Ct inclusions

InaC is required for Ct to assemble F-actin scaffolds and to reposition Golgi mini stacks around the periphery of the inclusion membrane (*Kokes et al., 2015*; *Wesolowski et al., 2017*; *Haines et al., 2021*). Because Cdu1 regulates InaC levels, we predicted that *cdu1* mutants would phenocopy *inaC* mutants. We quantified the number of inclusions surrounded by F-actin cages at 40 hpi. In cells infected with WT L2 (parental strain of M407 (*inaC* null), *Nguyen and Valdivia, 2012*; *Kokes et al., 2015*), approximately 25% of inclusions were surrounded by F-actin, consistent with previous observations (*Chin et al., 2012*; *Kokes et al., 2015*; *Figure 6A and B*, and *Figure 6—figure supplement 1*). The number of inclusions surrounded by F-actin decreased to approximately 7% in cells infected with an *inaC* null strain (M407) and increased to approximately 49% in HeLa cells infected with an *inaC* null strain (M407) complemented with wild type InaC (*Figure 6A and B*, and *Figure 6—figure supplement 1*). Cells infected with *cdu1*::GII mutants transformed with an empty plasmid or expressing Cdu1$^{C345A}$-Flag (DUB- Act-) and Cdu1$^{K268E}$-Flag (Act-) resulted in approximately 8%, 13%, and 10% of of F-actin positive inclusions respectively (*Figure 6A and B*, and *Figure 6—figure supplement 1*). In contrast, cells infected with *cdu1*::GII mutants expressing Cdu1-Flag and Cdu1$^{I225A}$-Flag (DUB-) led to a marked increase in F-actin inclusions (approximately 52% and 46%, respectively) (*Figure 6A and B*, and *Figure 6—figure supplement 1*). From these observations we conclude that the acetylase activity of Cdu1 is required for Ct to promote assembly of F-actin around the Ct inclusion likely through the stabilization of InaC.

We next quantified Golgi dispersal in infected HeLa cells at 24 hpi, a process that is also dependent on InaC (*Kokes et al., 2015*; *Wesolowski et al., 2017*). In HeLa cells infected with an *inaC* null strain (M407) Golgi dispersal was limited to approximately 26% of the Ct inclusion perimeter. In contrast, cells infected with either its parental WT L2 or with an *inaC* null strain (M407) complemented with wild type InaC, the Golgi is dispersed around 45% of the inclusion perimeter (*Figure 6C and D*, and *Figure 6—figure supplement 2*). Similarly, Golgi dispersal around inclusions during infection with WT L2 and in *cdu1*::GII mutants expressing wild type Cdu1-Flag or Cdu1$^{I225A}$-Flag (DUB-) was approximately 43%, 41%, and 43%, respectively (*Figure 6C and D*, and *Figure 6—figure supplement 2*). In HeLa cells infected with *cdu1*::GII and *cdu1*::GII strains expressing Cdu1$^{C345A}$-Flag (DUB- Act-), and Cdu1$^{K268E}$-Flag (Act-), Golgi repositioning was restricted to approximately 24%, 23%, and 23% of inclusion perimeters, respectively (*Figure 6C and D*, and *Figure 6—figure supplement 2*). These results confirm that both InaC and Cdu1 are required for efficient repositioning of the Golgi around the Ct inclusion as previously reported (*Kokes et al., 2015*; *Wesolowski et al., 2017*; *Pruneda et al., 2018*; *Auer et al., 2020*) and that this process is independent of Cdu1's DUB activity but requires its acetylase activity. Moreover, our results suggest that Cdu1 promotes Golgi repositioning by protecting InaC-mediated redistribution of the Golgi around the Ct inclusion.

CTL0480 promotes recruitment of the myosin phosphatase subunit MYPT1 to the inclusion membrane where it regulates the extrusion of intact inclusions from host cells (*Lutter et al., 2013*; *Shaw et al., 2018*). Consistent with the gradual loss of CTL0480 from inclusions in cells infected with the *cdu1*::GII strain starting at 36 hpi (*Figure 3E* and *Figure 3—figure supplement 1*) we also observed a complete loss of MYPT1 recruitment to inclusions by 48 hpi (*Figure 6E and F*).

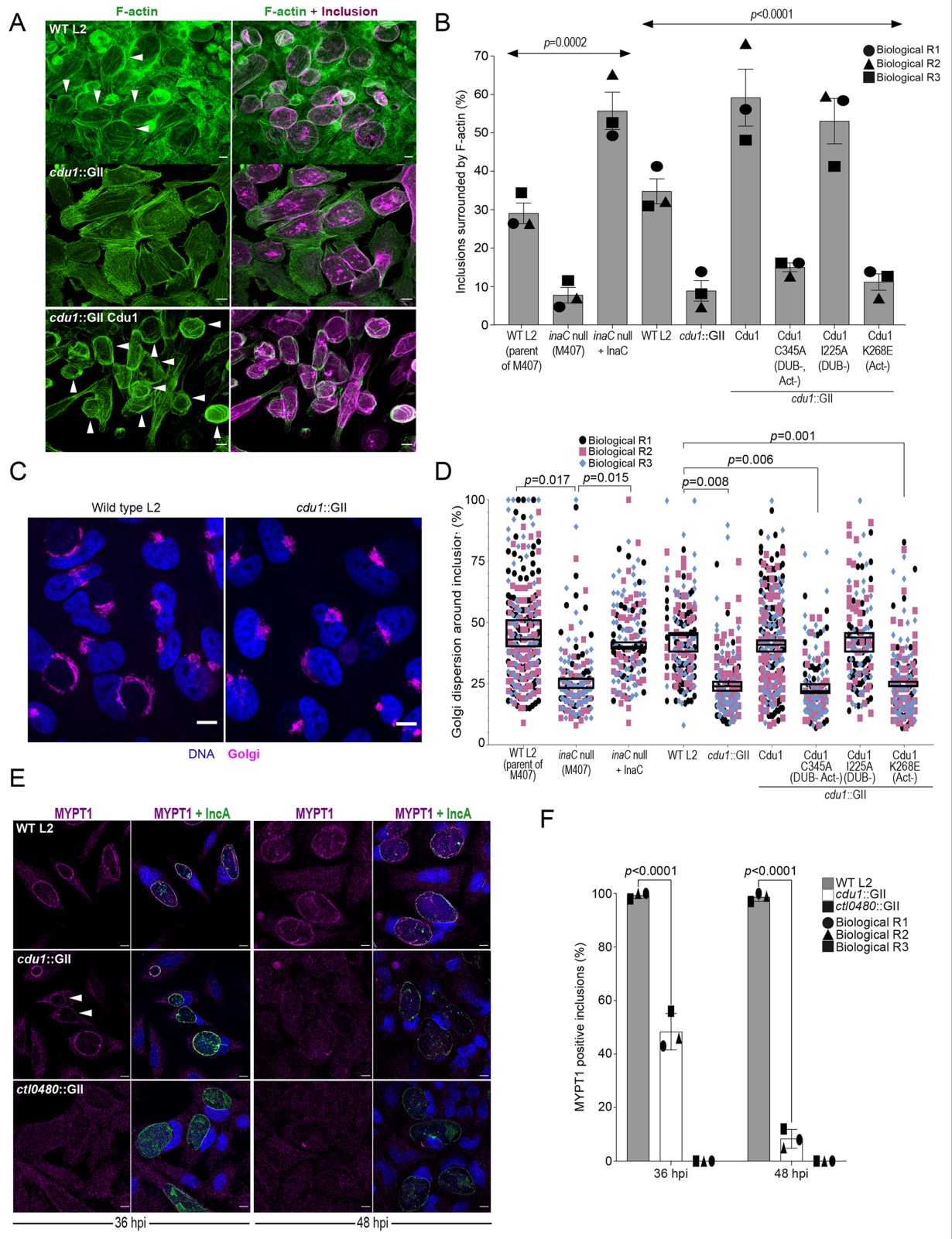

**Figure 6.** Cdu1 is required for assembly of F-actin, Golgi ministack repositioning, and MYPT1 recruitment to the inclusion. (**A**) Examples of representative images of F-actin (arrowheads) (green, Alexa Fluor Phalloidin) assembled around the Ct inclusion (magenta, anti Cdu1 and Cap1 staining) in HeLa cells infected for 40 hr. Representative images for each strain analyzed can be found in *Figure 6—figure supplement 1*. (**B**) Quantification of Ct inclusion surrounded by F-actin normalized to the total number of inclusions analyzed during infection of HeLa cells at 40 hpi. Quantification of

*Figure 6 continued on next page*

*Figure 6 continued*

surrounding F-actin were obtained from inclusions imaged in six fields across three independent biological replicates. p values were determined by one-way ANOVAs with a Student-Newman-Keuls post hoc test. Strains used: WT L2 (Rif-R 434 Bu, parent of M407), M407 (*inaC* null strain) p2TK2, M407 p2TK2-InaC, WT L2 (434 Bu) pBOMB, *cdu1*::GII pBOMB, *cdu1*::GII pBOMB-Cdu1 Flag, *cdu1*::GII pBOMB-Cdu1C345A Flag, *cdu1*::GII pBOMB-Cdu1I225A Flag, and *cdu1*::GII pBOMB-Cdu1K268E Flag. For cells infected with the strains WT CT (parent of M407), *inaC* null, *inaC* null + InaC, WT L2, *cdu1*::GII, and *cdu1*::GII strains expressing WT Cdu1 and the variants C345A, I225A and K268E, n (across 3 replicates) = 259, 708, 780, 477, 538, 496, 370, 438, and 472 respectively. Error bars depict standard deviation.(**C**) Sample representative images of Golgi (anti GM130 staining, magenta) around Ct inclusions in HeLa cells infected for 24 hours. Representative images for each strain analyzed can be found in *Figure 6—figure supplement 2*. (**D**) Quantification of Golgi dispersal around the Ct inclusion during infection of HeLa cells for 24 hpi. The length of Golgi dispersed around each Ct inclusion imaged was measured and normalized to the perimeter length of each inclusion (% Golgi dispersion around the inclusion). Golgi dispersal around Ct inclusions was quantified from inclusions imaged in six fields across three independent biological replicates. p values were determined by a student paired t-test. Strains analyzed were the same ones as mentioned in (**B**). For cells infected with the strains WT CT (parent of M407), *inaC* null, *inaC* null + InaC, WT L2, *cdu1*::GII, and *cdu1*::GII strains expressing WT Cdu1 and the variants C345A, I225A and K268E, n (across 3 replicates) = 351, 187, 147, 169, 198, 284,187, 143, and 130 respectively. Error bars depict standard deviation. (**E**) Representative images of MYPT1 (magenta) at Ct inclusions (green, anti-IncA staining). Arrowheads represent *cdu1* null inclusions with low MYPT1 signal. DNA stained with Hoechst is shown in blue in panels C and E. Scale bar: 10 µm. (**F**) Quantification of MYPT1 at Ct inclusions as shown in (**E**). Representative images in (**E**) and quantification of MYPT1 recruitment in (**F**) were obtained from inclusions imaged in six fields across three independent replicates. Error bars depict standard deviation. p values were determined by a student paired t-test. For cells infected with WT Ct n (across 3 replicates) = 187 inclusions at 36 hpi and n=204 inclusions at 48 hpi. For cells infected with *cdu1*::GII n (across 3 replicates) = 202 inclusions at 36 hpi and n=168 at 48 hpi. For cells infected with *ctl0480*::GII n (across 3 replicates) = 175 inclusions at 36 hpi and n=203 inclusions at 48 hpi.

The online version of this article includes the following source data and figure supplement(s) for figure 6:

**Source data 1.** Excel file containing numerical data for quantifying the number of inclusions surrounded by filamentous actin in HeLa cells infected with WT CT, *inaC* null, *inaC* null complemented with InaC, *cdu1*::GII, and *cdu1*::GII strains expressing Cdu1, Cdu1 C345A, I225A, and K268E variants (*Figure 6B*).

**Source data 2.** Excel files containing numerical data for quantifying the number of inclusions surrounded by dispersed Golgi mini stacks in HeLa cells infected with WT CT, *inaC* null, *inaC* null complemented with InaC, *cdu1*::GII, and *cdu1*::GII strains expressing Cdu1, Cdu1 C345A, I225A, and K268E variants (*Figure 6D*).

**Source data 3.** Excel files with numerical data for quantifying the localization of MYPT1 to WT and *cdu1*::GII inclusions (*Figure 6F*).

**Figure supplement 1.** The DUB activity of Cdu1 is not required for assembly of F-actin around the Ct inclusion.

**Figure supplement 2.** The DUB activity of Cdu1 is not required for Golgi ministack repositioning around the Ct inclusion.

## Cdu1, InaC, IpaM, and CTL0480 are required for optimal extrusion of Ct from host cells

In the absence of Cdu1, the levels of InaC, CTL0480, and IpaM decreased late in infection (36 hpi and 48 hpi, *Figures 3 and 4*) suggesting that a prominent role of Cdu1 is to protect these Incs from degradation late in infection. At the end of its developmental cycle, *Chlamydia* exits host cells by promoting cellular lysis or by extrusion of intact inclusions (*Hybiske and Stephens, 2007*). Ct host cell exit by extrusion is an active process requiring a remodeling of the actin cytoskeleton and the function of Inc proteins (*Hybiske and Stephens, 2007*; *Chin et al., 2012*; *Lutter et al., 2013*; *Shaw et al., 2018*; *Nguyen et al., 2018*). CTL0480 recruits MYPT1 (an inhibitor of Myosin II motor complexes) to the inclusion membrane which prevents premature extrusion of Ct inclusions and loss of CTL0480 leads to increased rates of extrusion by Ct from infected HeLa cells (*Lutter et al., 2013*; *Shaw et al., 2018*; *Figure 7*). Actin polymerization is also required for Ct extrusion (*Hybiske and Stephens, 2007*; *Chin et al., 2012*) suggesting that InaC dependent recruitment of F-actin to the inclusion may also contribute to optimal Ct extrusion. IpaM localizes to microdomains in the inclusion membrane that are proposed to function as foci for extrusion (*Nguyen et al., 2018*). Based on these observations, we postulated that Cdu1-mediated protection of CTL0480, InaC, and IpaM regulates the extrusion of Ct inclusions. We quantified the number of extrusions released from infected HeLa cells at 52 hpi and observed a 60% reduction in the number of extrusions in HeLa cells infected with the *cdu1*::GII strain relative to cells infected with WT L2 (*Figure 7A and B*). Complementation of *cdu1*::GII with either wild type Cdu1-Flag or Cdu1I225A-Flag (DUB-) restored extrusion production to near wild type levels. In contrast, HeLa cells infected with *cdu1*::GII mutants expressing Cdu1K268E-Flag (Act-), or *inaC* (*inaC*::GII, *Wesolowski et al., 2017*) and *ipaM* (*ipaM*::GII, *Meier et al., 2023*) null strains led to a 42%, 75%, and 58% reduction in extrusion production respectively (*Figure 7A and B*). The decrease in the number of extruded inclusions by these strains was not attributed to defects in inclusion biogenesis

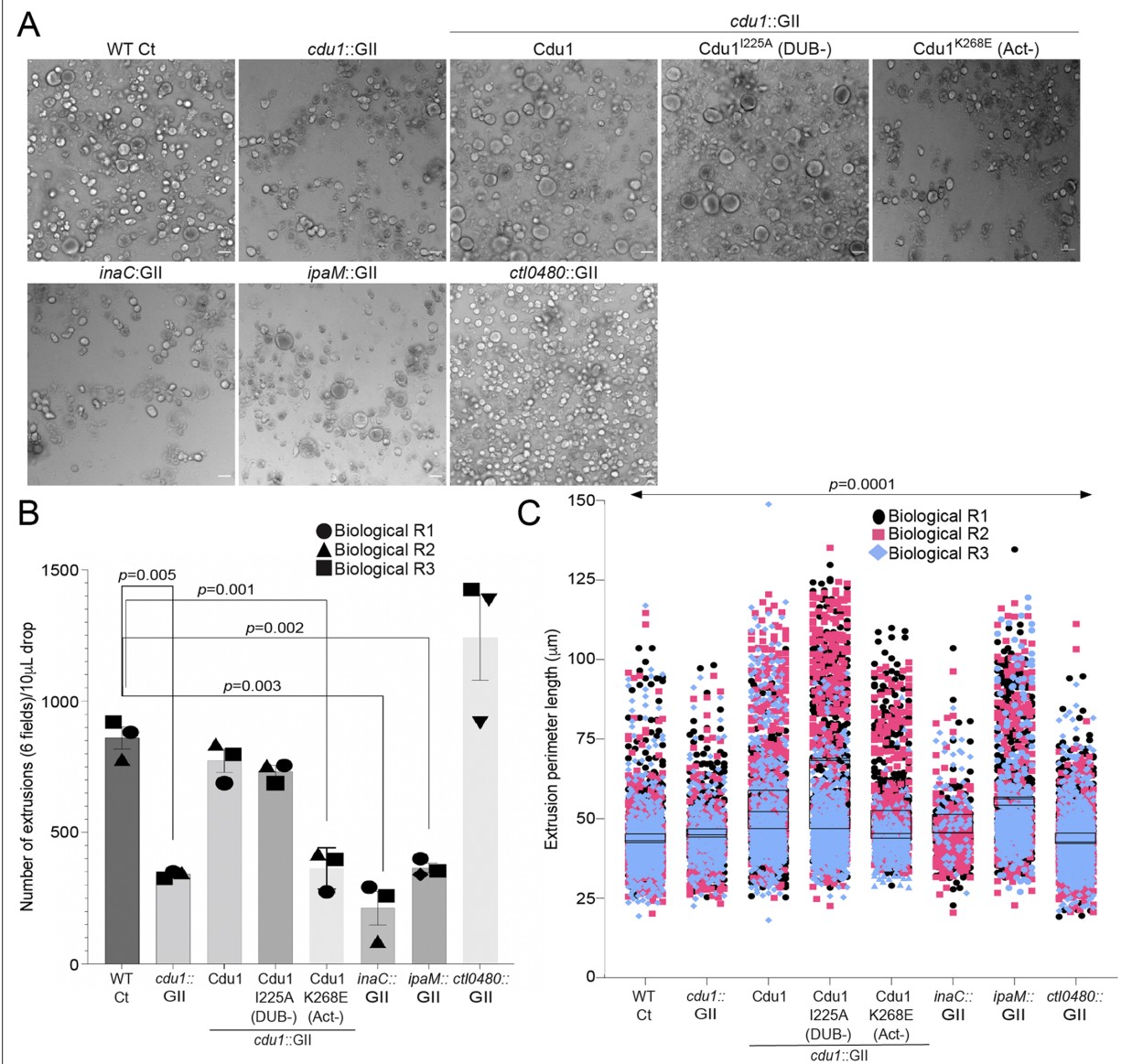

**Figure 7.** Cdu1, InaC, and IpaM are required for optimal extrusion of Ct inclusions from HeLa cells. (**A**) Representative images of extrusions isolated from HeLa cell monolayers infected with Ct strains for 52 hr. Scale bar: 200 µm (**B**) Quantification of the number of extruded inclusions produced by infected HeLa cell monolayers. p values were determined by a student paired t-test. (**C**) Quantification of the size of extruded inclusions quantified in (**B**). Extruded inclusions varied in size among cells infected with wild type L2 (average:43 µm), *ipaM* mutants (average: 56 µm) and *cdu1* mutants complemented wild type Cdu1 (average: 52 µm) and Cdu1$^{I225A}$ (DUB-) (average: 60 µm). *p* values were determined by one-way ANOVAs with a Student-Newman-Keuls post hoc test. Representative images in (**A**) and quantification of extrusion number in (**B**) and extrusion size in (**C**) are based on images obtained from six fields across three independent biological replicates. For extrusions isolated from Hela cells infected with WT CT, *cdu1*::GII, *inaC*::GII, *ipaM*::GII, *ctl0480*::GII, and *cdu1*::GII strains expressing WT Cdu1 and the variants I225A and K268E, n (across 3 replicates) = 2,580, 1,025, 637, 1,194, 3,726, 2,321, 2,198, and 1,091 respectively. Error bars depict standard deviation.

The online version of this article includes the following source data and figure supplement(s) for figure 7:

**Source data 1.** Excel files containing numerical data for quantifying the number of extrusions produced and the size of extrusions in WT CT, *cdu1*::GII strains, and *cdu1*::GII strains expressing Cdu1, I225A, and K268E variants from infected HeLa cells (***Figure 7B and C***).

**Figure supplement 1.** The number of inclusions and the size of inclusions in *cdu1* null, *inaC* null, *ipaM* null, and *ctl0480* null strains are similar across each strain.

**Figure supplement 1—source data 1.** Excel files with numerical data for quantifying the number of inclusions produced and inclusion size in WT CT, *cdu1*::GII strains, and *cdu1*::GII strains expressing Cdu1, I225A, and K268E variants in infected HeLa cells (panels B and C).

as they produced a comparable number of inclusions at 48 hpi relative to cells infected with WT L2 (*Figure 7—figure supplement 1*). Consequently, we infer that InaC, IpaM, and Cdu1 collectively contribute to the promotion of optimal extrusion of Ct inclusions from host cells, with Cdu1 playing a central regulatory role by protecting these effectors from degradation.

In contrast, infection of HeLa cells with a *ctl0480::GII* mutant strain led to an increase in the number of extruded inclusions as previously observed (*Shaw et al., 2018*; *Figure 7A and B*). Therefore, even though the Cdu1-mediated protection of InaC and IpaM is important for the extrusion of inclusions and *cdu1* mutants phenocopy the loss of InaC and IpaM, the phenotypic similarities do not extend to the increased number of extruded inclusions observed in cells infected with the *ctl0480*::GII mutant strain (*Figure 7A and B*). We infer from these observations that functions for both InaC and IpaM in the extrusion of inclusions are epistatic to CTL0480. Extruded inclusions produced during infection of HeLa cells also varied in size with an average diameter of 40 μm (*Figure 7A and C*). Interestingly, the loss of IpaM and over expression of Cdu1-Flag and Cdu1$^{I225A}$-Flag (DUB-) shifted the size distribution of extrusions toward larger extrusions (*Figure 7A and C*) suggesting that Ct regulates the size of extruded inclusions through Cdu1.

## Discussion

Several *Chlamydia* Inc proteins regulate interactions between the pathogenic vacuole and the host cytoskeleton, organelles, and vesicular trafficking pathways. These Inc proteins also modulate host cell death programs and promote *Chlamydia* exit from host cells (reviewed in *Bugalhão and Mota, 2019*). Given the central roles that Incs play in promoting *Chlamydia* intracellular infection, it is not surprising that they are targeted for inactivation by host cellular defenses. In response, *Chlamydia* has evolved mechanisms to protect Incs. In this study, we show that the acetylase activity of the effector

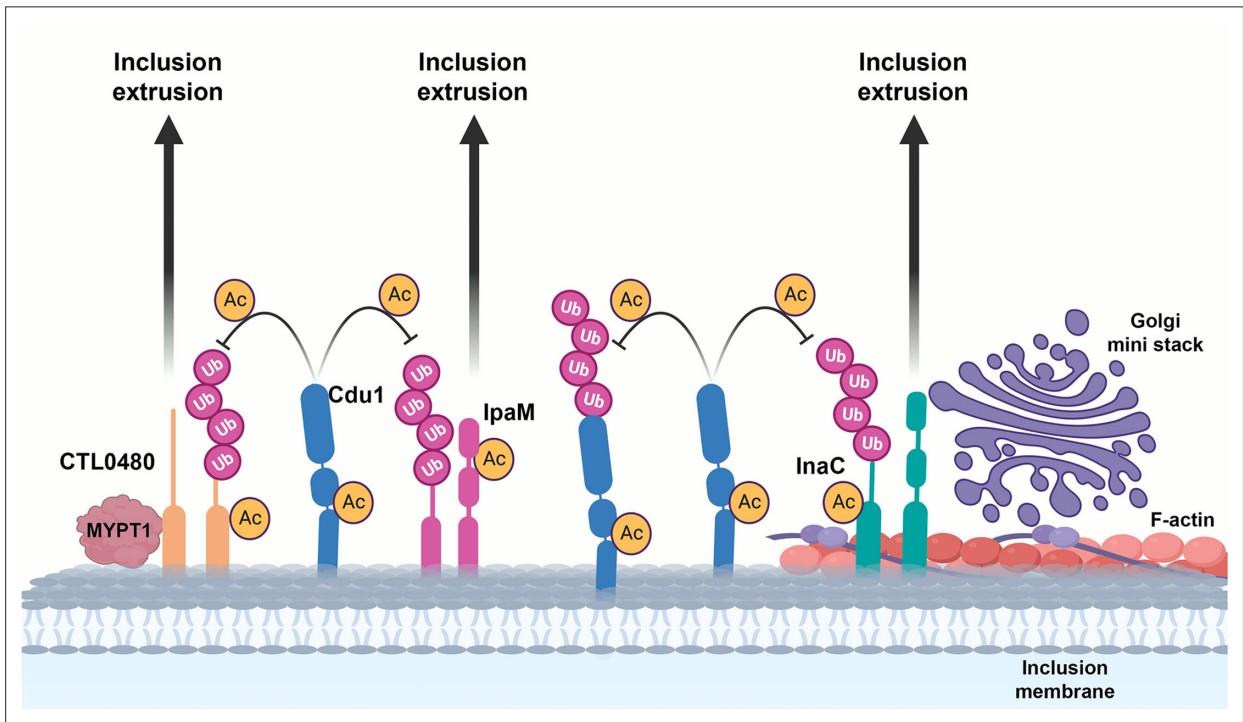

**Figure 8.** A model for acetylation mediated protection of the Inc proteins InaC, IpaM, and Ctl0480 from degradation. The cellular Ub machinery targets *C. trachomatis* effectors, including the Inc proteins InaC, IpaM, and CTL0480 for ubiquitination and subsequent protein degradation. *C. trachomatis* counters such defense mechanisms by translocating Cdu1 which protect itself and all three Inc proteins from being targeted for ubiquitination and degradation through its acetylase activity. Cdu1 protects InaC and enables the recruitment of F-actin scaffolds and Golgi ministacks to the inclusion perimeter and CTL0480 to recruit the Myosin II regulator MYPT1. All three inclusion proteins and Cdu1 promote extrusion and dissemination of *C. trachomatis* inclusions late in infection.

Cdu1 protects itself and three Inc proteins; InaC, IpaM, and CTL0480, from ubiquitination and degradation (*Figure 8*). Interestingly, all three Inc proteins play prominent roles in regulating the extrusion of inclusions from host cells (*Figure 7*). Observations that the encapsulation of *Chlamydia* within an extruded inclusion enhances survival of *Chlamydia* within macrophages (*Zuck et al., 2017*) together with the broad conservation of extrusion as an exit strategy among *Chlamydia* (*Zuck et al., 2016*) suggests that this mechanism is important for *Chlamydia* pathogenesis. Notably, a *cdu1* mutant strain (*cdu1*:: Tn, *Fischer et al., 2017*) displays reduced bacterial loads in a murine model of upper genital tract infections (*Fischer et al., 2017*). While neither this strain nor our *cdu1*::GII mutant strain shows evident growth impairments during infection of HeLa cells (data not shown), the observed reduction in bacterial load in the absence of Cdu1 in animal models of infection could potentially stem from defects in extrusion production or from perturbations in Cdu1-dependent regulation of extrusion size. Thus, targeting Inc proteins that regulate extrusion for Ub-mediated destruction may be advantageous for the host. For instance, targeting InaC for degradation would limit F-actin dependent extrusions (*Hybiske and Stephens, 2007*; *Chin et al., 2012*) and InaC-dependent microtubule scaffolds around the inclusion (*Wesolowski et al., 2017*; *Haines et al., 2021*). CTL0480 functions as an inhibitor of extrusions through its role in modulating the activity of myosin light chain 2 (MLC$_2$) (*Lutter et al., 2013*; *Shaw et al., 2018*). IpaM localizes to specialized microdomains in the inclusion membrane which are also sites of enrichment for over 9 inclusion membrane proteins including Ctl0480 and MrcA, both of which are required for *Chlamydia* extrusion (*Mital et al., 2010*; *Lutter et al., 2013*; *Nguyen et al., 2018*). We also find that the loss of IpaM shifted the size distribution of extrusions towards larger inclusions (*Figure 7*). We speculate that heterogeneity in the size of extrusions might facilitate uptake of some extrusions by innate immune cells at infected mucosal sites to promote *Chlamydia* LGV dissemination to distal sites in the genital tract and avoid clearance of *Chlamydia* by other immune cells (*Zuck et al., 2017*).

Effectors that modulate the activity of other translocated effectors are referred to as 'metaeffectors', a term coined by Kubori and colleagues after observing that the *L. pneumophila* effector LubX which functions as an E3 ligase, ubiquitinates the translocated effector SidH leading to its degradation (*Kubori et al., 2010*). Several other effector-metaeffector interactions have been described in *L. pneumophila*, *Salmonella enterica*, and *Brucella abortus* which regulate the activity of other effectors either directly or indirectly by modifying the same host target or cellular process (*Kubori et al., 2010*; *Neunuebel et al., 2011*; *Jeong et al., 2015*; *Urbanus et al., 2016*; *Smith et al., 2020*; *Iyer and Das, 2021*). In this context, we propose that Cdu1 functions as a metaeffector in Ct to protect multiple effectors. We also observed that Cdu1 interactions with InaC, IpaM, and CTL0480 likely occur independently from each other and that the kinetics of degradation in the absence of Cdu1 varies for each Inc (*Figure 3*).

Our findings indicate that the DUB activity of Cdu1 was not required to protect InaC, IpaM, and CTL0480 from ubiquitination. Instead we find that Cdu1's lysine acetylase activity is required to protect these Inc proteins and Cdu1 itself from ubiquitination. Indeed, we found that all three Incs and Cdu1 are acetylated at lysines in infected cells. However, we were unable to determine if lysine acetylation in all four proteins was dependent on Cdu1s' Act activity or if these PTMs are protective. Why the DUB activity of Cdu1 is unable to compensate for loss of its Act activity remains unknown. It is possible that Cdu1, like other DUBs, is regulated by PTMs (*Komander et al., 2009a*). For instance, phosphorylation of human CYLD inhibits its DUB activity towards TRAF2 while phosphorylation of human USP8 inhibits its DUB activity toward EGFR (*Reiley et al., 2005*; *Mizuno et al., 2007*). Mass spectrometry analysis of immunoprecipitated Flag tagged Cdu1 expressed in Ct revealed that Cdu1 is phosphorylated at multiple serine and threonine residues (*Figure 4H*) as previously suggested (*Zadora et al., 2019*). We identified three PX(S/T)P MAPK phosphorylation consensus sequence motifs in the proline rich domain (PRD) of Cdu1, suggesting that MAPKs may regulate the DUB activity of Cdu1.

Cdu1 homologs are found in multiple *Chlamydia* species including *C. trachomatis*, *C. muridarum*, *C. suis*, *C. psitacci*, *C. abortus*, *C. caviae*, and *C. felis* but is notably absent in the genomes of *C. pneumoniae* and *C. pecorum*. The acquisition of a second deubiquitinase paralog (Cdu2) has also occurred in *C. trachomatis*, *C. muridarum*, and *C. suis*. In the genomes of all three species, *cdu2* resides directly adjacent to *cdu1*; an arrangement that presumptively arose from a gene duplication event. Cdu2 is a dedicated ULP with deubiquitinating and deneddylating activities (*Misaghi et al., 2006*; *Pruneda et al., 2016*). New evidence suggest that both paralogues might not be functionally

redundant. The crystal structure of Cdu2 has revealed differences in residues involved in substrate recognition between Cdu1 and Cdu2 and that each paralog might recognize polyUb chains differently (*Hausman et al., 2020*). The processivity rates for removal of terminal Ub from polyUb chains also differs between both isopeptidases with Cdu2 exhibiting limited trimming of polyUb as compared to Cdu1 (*Hausman et al., 2020*). Moreover, Cdu2 lacks the proline rich domain found in Cdu1 which might be important for regulation of Cdu1 enzymatic activity. The presence of Cdu2 might also explain the low incidence of human and Ct proteins that were differentially ubiquitinated in the absence of Cdu1 (*Figure 1*). Whereas several *Chlamydia* species have acquired either one or two deubiquitinase paralogs, both *C. pneumoniae* and *C. pecorum* have not. Instead, both species have acquired an unrelated deubiquitinase (*Chla*OTU) belonging to the OTU family of proteases (*Makarova et al., 2000*; *Furtado et al., 2013*). Curiously, *Chla*OTU is also found in *C. psitacci*, *C. abortus*, *C. caviae*, and *C. felis* all of which encode only Cdu1 and is absent in *C. trachomatis*, *C. muridarum*, and *C. suis*, all of which encode Cdu1 and Cdu2. It is noteworthy that *Chlamydia* species have independently acquired deubiquitinases multiple times (Cdu1, Cdu2, *Chla*OTU) and that some of these deubiquitinases have evolved into moonlighting enzymes reflecting the diverse strategies adopted by pathogenic *Chlamydia* as they adapt to their particular niche.

# Materials and methods

## Key resource table
See appendix 1.

## Resource availability
### Materials availability
All newly generated materials associated with this study will be freely available upon request.

## Experimental model and subject details
### Cell lines
Vero (CCL-81; RRID:CVCL_0030), HeLa (CCL-2; RRID:CVCL_0059), HEK293T (CRL-3216; RRID:CVCL_0063), and A549 (CCL-185; RRID:CVCL_0023) cells were purchased from ATCC and cultured in High Glucose Dulbecco's Modified Eagle's Medium supplemented with L-glutamine, sodium pyruvate (DMEM; Gibco) and 10% fetal bovine serum (FBS; Sigma-Aldrich). Cells were grown at 37 °C in a 5% CO2 humidified incubator. Vero, HeLa, and HEK293T cells were derived from females while A549 cells were derived from a male. All four cell lines have been authenticated by the Duke Cell Culture and DNA analysis facility and routinely tested for the presence of Mycoplasma.

### *Chlamydia* strains and propagation
*Chlamydia* strains used in this study are listed in the Key Resources Table. Ct strains were propagated in Vero cells and harvested by osmotic lysis at 48 hpi. Following lysis extracts were sonicated and bacteria pelleted by centrifugation at 21,000 x *g*. Bacteria were resuspended in SPG storage buffer (75 g/L sucrose, 0.5 g/L KH$_4$HPO$_4$, 1.2 g/L Na$_2$HPO$_4$, 0.72 g/L glutamic acid, pH 7.5) and stored as single use aliquots at –80 °C.

## Method etails
### *Chlamydia* infections
*Chlamydia* infections were synchronized by centrifugation (2500 x *g* for 30 min at 10 °C) onto HeLa cell monolayers and incubated for the indicated times. Co-infections were performed by infecting HeLa cell monolayers at a 1:1 ratio using MOIs of 2 for each co-infecting strain.

### Insertional mutagenesis of CTL0247 (*cdu1*)
Primer sequences for TargeTron mediated mutagenesis of the LGV L2 434 Bu *cdu1* (CTL0247) ORF were designed at the TARGETRONICS, LLC web portal (https://www.targetrons.com/). IBS1/2, EBS1/delta, and EBS2 primers (primer sequences are listed in *Supplementary file 10*) were used in a PCR reaction to generate homing sequences for TargeTron integration between nucleotides 635 and 636

of the *cdu1* ORF using a TargeTron gene knockout system (Sigma-Aldrich; TA0100) according to the manufacturer instructions. Homing sequences were gel purified, digested with HindIII and BsrGI, and ligated into HindIII and BsrGI digested pDFTT3-*aadA* (*Lowden et al., 2015*). Ligations were transformed into *E. coli* DH5α, clones isolated, and *cdu1* redirected pDFTT3-aadA plasmids identified by restriction digest and verified by Sanger sequencing (Eton Bioscience) using a T7-promoter specific primer. The resulting plasmid was transformed into a *C. trachomatis* LGV L2 434 Bu strain and transformants selected with 150 μg/mL spectinomycin and plaque purified as previously described (*Kędzior and Bastidas, 2019*). Insertion of the GII *aadA* intron at the *cdu1* locus was verified by PCR analysis (S. *Figure 1—figure supplement 1*) using primers that amplify amplicons spanning the *cdu1*::GII 5' (RBP409 and RBP436) and 3' (RBP468 and RBP118) junctions, the *cdu1* CDS (RBP409 and RBP118), and the *aadA* CDS (RBP512 and RBP513). Primer sequences are listed in *Supplementary file 10*. Loss of Cdu1 protein was verified by western blot and indirect immunofluorescence analysis (S. *Figure 2A and B*).

## Analysis of *cdu1* and *cdu2* transcription by RT-PCR

Confluent HeLa cell monolayers ($2.9 \times 10^6$ cells/infection) were infected with wild type L2 434 Bu or L2 *cdu1*::GII *aadA* strains. At 24 hpi, total RNA was isolated with a Qiagen RNeasy kit (Qiagen; 74004) according to the manufacturer instructions. Total RNA was treated twice with DNAse I (NEB; M0303S) and used for cDNA synthesis using a SuperScript IV Reverse Transcriptase kit (Thermo Fisher Scientific; 18090010). cDNAs synthesized with and without reverse transcriptase were used as templates for PCR analysis (S. *Figure 2C*) using primers that amplify amplicons spanning the *cdu*1 (CTL0247_F and CTL0247_R) and *cdu2* ORFs (CTL0246_F and CTL0247_R), and the intergenic junction between the *cdu1* and *cdu2* ORFs (CTL0246-0247_F and CTL0246-0247_R). Primer sequences can be found in *Supplementary file 10*.

## TUBE1 based global ubiquitin profiling

Confluent HeLa cell monolayers ($5.04 \times 10^6$ cells/infection) were mock infected or separately infected with WT LGV L2 434 Bu or a L2 *cdu1*::GII *aadA* strain at MOIs of 3. At 24 hpi cells were collected and spun down (700 x *g* for 10 min), frozen at –80 °C and shipped on dry ice to LifeSensors (Malvern, PA) for quantitative TUBE1-based Mass Spectrometry Analysis. Cell pellets from three independent biological replicates were sent to LifeSensors for analysis. Cell were subsequently lysed in lysis buffer (50 mM Tris-HCL, pH 7.5, 150 mM NaCl, 2 mM EDTA, 1% NP-40, 10% glycerol, 1% Sodium Deoxycholate) supplemented with a protease inhibitor cocktail, the DUB inhibitor PR-619 (Sigma-Aldrich; SML0430), and the proteasomal inhibitor MG-132 (Sigma-Aldrich; 474791). Lysates were clarified by high-speed centrifugation (14,000 x *g*, 10 min, 4 °C) and supernatants containing 5 mg of protein were equilibrated with magnetic TUBE-1 (LifeSensors; UM401M) and incubated overnight at 4 °C under rotation. TUBEs were isolated with a magnetic stand and washed sequentially with PBST and TUBE wash buffer. Poly-ubiquitinated and associated proteins were eluted with TUBE elution buffer. Eluted supernatants were neutralized with neutralization buffer and loaded onto SDS-gels and run until SDS buffer reached 0.5 cm into the gel. Gels were stained with Coomassie Blue and lanes excised, reduced with TCEP, alkylated with iodoacetamide, and digested with Trypsin (Trypsin Gold, Mass Spectrometry Grade) (Promega; V5280). Tryptic digests were analyzed using a 150 min LC run on a Thermo Scientific Q Exactive HF Orbitrap LC-MS/MS system. MS data was searched against the UniProt human database (UniProt; Proteome ID: UP000005640) and the *Chlamydia trachomatis* L2 434 Bu reference database (NCBI:txid47472) using MaxQuant 1.6.2.3 (*Cox and Mann, 2018*). Proteins, peptides, and site identification was set to a false discovery rate of 1%. N-terminal acetylation, Met oxidation, and diGly remnant on lysine residues was also identified. All peptides and proteins identified can be found in *Supplementary file 1*. The intensities (sum of all peptide MS peak areas for a protein or ubiquitinated peptide) for each protein and ubiquitinated peptide across all three biological replicates were used to determine mean intensities and to calculate *p*-values based on one-way student t-tests. Volcano plots of mean intensities vs. p-values were generated with VolcaNoseR (*Goedhart and Luijsterburg, 2020*) and used to identify significantly enriched proteins and ubiquitinated peptides. Data used to generate each Volcano plot can be found in *Supplementary file 2*. Pathway enrichment analysis was performed with Metascape (*Zhou et al., 2019*) and DAVID bioinformatic resources (*Huang et al., 2009a*; *Huang et al., 2009b*).

## Inhibitors, antibodies, western blots, and densitometry analysis

MG-132 (25 µM) (Sigma-Aldrich; 474791) was added to infected cell monolayers 5 h prior to extract preparations. Recombinant Cdu1 protein (LGV L2 434 Bu, amino acids 71–401) was generated as previously described (*Pruneda et al., 2018*) and kindly provided by Jonathan Pruneda (Oregon Health and Science University, OR). Recombinant Cdu1 protein was used to generate antibodies in immunized New Zealand White rabbits. Cdu1 antisera was pre-adsorbed with crude cell extracts from HeLa cells infected with a *cdu1*::GII *aadA* strain. Pre-adsorbed antisera was used for western blot analysis at a 1:500 dilution in a solution containing 5% BSA supplemented with crude extracts from HeLa cells infected with a *cdu1*::GII *aadA* strain (0.1 mg/mL total protein). Antibodies, antibody dilutions, and antibody diluents used in this study are listed in *Supplementary file 9*. For western blot analysis, lysates from infected HeLa cell monolayers (2.4x10$^6$ cells) were prepared by incubating cell monolayers with boiling hot 1% SDS lysis buffer (1% SDS, 100 mM NaCl, 50 mM Tris, pH 7.5). Lysates were collected, briefly sonicated, and total protein concentration measured with a DC Protein Assay Kit (BIO-RAD; 5000111). Eight µg (Slc1 and alpha Tubulin blots) and 25 µg of total protein lysates (all other blots) were loaded onto 4–15% Mini-PROTEAN and TGX Stain Free Protein Gels (Bio-Rad; 4568084), transferred to PVDF membranes (Bio-Rad; 1620177), blocked with 5% Milk/TBSt, and incubated with primary antibodies overnight at 4 °C. Protein signals were detected with Goat anti mouse (H+L) IgG (Thermo Fisher Scientific; 31430) or Goat anti-rabbit (H+L) IgG HRP (Thermo Fisher Scientific; 31460) conjugated secondary antibodies (1:1000 in 5% Milk/TBSt) and SuperSignal West Femto HRP substrate (ThermoFisher scientific; 34096). Antibody-bound membranes were imaged with a LI-COR Odyssey Fc Imager (LI-COR, Inc). Varying amounts of protein extracts were used to determine the linear range of detection for InaC, IpaM, and Slc1 antibodies prior to quantification of western blot images (data not shown). Protein bands were quantified using western blot densitometry analysis with LI-COR Image Studio Software (LI-COR, Inc). InaC and IpaM densitometry measurments were normalized to corresponding Slc1 densitometry measurments.

## Immunofluorescence microscopy

HeLa cells were grown on coverslips to 50% confluency (0.1x10$^5$ cells) and infected at MOIs of 0.6. At indicated times, infected cells were separately fixed with ice cold Methanol or with warm PBS containing 4% formaldehyde for 20 min. After fixative removal, cells were washed with PBS and formaldehyde fixed cells were incubated either in 5% BSA/PBS supplemented with 0.1% Triton X-100 or in 5% BSA/PBS supplemented with 0.05% Saponin for 30 min with gentle rocking. Following washing with PBS, Methanol fixed cells were incubated with primary antibodies diluted in 5% BSA/PBS and formaldehyde fixed cells were incubated with primary antibodies diluted in 5% BSA/PBS supplemented with 0.1% Triton X-100 or 0.05% Saponin for 1 hr with gentle rocking. Dilutions for each antibody used can be found in *Supplementary file 9*. Methanol fixed cells were washed with PBS and incubated with secondary antibodies diluted in 5% BSA/PBS and supplemented with Hoechst 33342 (2 µg/mL) (Thermo Fisher Scientific; H3570). Formaldehyde fixed cells were washed and incubated with 5% BSA/PBS supplemented with 0.1% Triton X-100 and Hoechst or 0.05% Saponin and Hoechst for 1 hr protected from light and with gentle rocking. For detection of F-actin, Phalloidin conjugated to Alexa Fluor 488 (1:5000) (Act-Stain 488 Phalloidin; Cytoskeleton Inc; PHDG1) was added for the last 20 min of incubation with the secondary antibodies. Coverslips were transferred to glass slides, mounted with 10 µL of Vectashield (Vector Labs; H-1000) and incubated over night at room temperature prior to imaging. Secondary antibodies used were goat anti-mouse (H+L) IgG (Thermo Fisher Scientific; A-11001 and A-21235) and goat anti-rabbit (H+L) IgG (Thermo Fisher Scientific; A-11008 and A-21244) conjugated to Alexa Fluor 488 and Alexa Fluor 647. All the antibodies used for indirect immunofluorescence analysis were analyzed under all three staining conditions (Methanol, Formaldehyde/Triton X-100, and Formaldehyde/Saponin).

Quantitative immunofluorescent microscopy for RNF213 was performed as previously described (*Walsh et al., 2022*). Briefly, A549 cells were grown on coverslips in 24-well plates to full confluency (~2 x 10$^5$ cells). Cells were infected with indicated *C. trachomatis* strains at an MOI of 2. At 3 hr post-infection, all cells were given fresh DMEM supplemented with L-tryptophan (100 µg/mL) with half of the wells given interferon-gamma (100 U/mL; Millipore, IF005). At 24 hr post-infection, cells were fixed with cold, 4% PFA in PBS for 20 min. Cells were permeabilized with ice-cold methanol for 1 min and blocked in PBS containing 5% BSA and 2.2% glycine for 30 min. Antibody incubations and microscope

slide mounting was performed as described. Samples were blinded using tape and imaged on a Zeiss Axio Observer.Z1 epifluorescent microscope. For each sample, at least 6 separate fields of view and 100 *Chlamydia* inclusions were captured, saved and further blinded using the ImageJ Blind Analysis Tool plugin. Quantification of the number of inclusions with RNF213 targeted to the inclusion membrane was performed using ImageJ (*Schneider et al., 2012*). Targeted inclusions were scored as having the indicated protein signal colocalize with >50% of the inclusion membrane signal (incA-positive antibody staining).

Representative images were acquired with an inverted confocal laser scanning microscope (Zeiss 880) equipped with an Airyscan detector (Hamamatsu) and with diode (405 nm), argon ion (488 nm), double solid-state (561 nm), and helium-neon (633 nm) lasers. Images were acquired with a 63 x C-Apochromatic NA 1.2 oil-objective (Zeiss). Images acquired in Airyscan mode were deconvoluted using automatic Airyscan processing in Zen software (Zeiss). Image acquisition was performed at the Light Microscopy Core Facility at Duke University. Images used for quantification were captured in an inverted microscope (Ti2-Nikon instruments) equipped with an ORCA Flash 4.0 V3 sCMOS camera (Hamamatsu) and a SOLA solid-state white light illuminator (Lumencro). Images were acquired using a 60 x Plan Apochromatic NA 1.40 oil objective. All images were opened with ImageJ (*Schneider et al., 2012*) and only linear adjustments were made to fluorescence intensity for the entire image. Images were exported as TIFFs and compiled with Adobe suite software (Illustrator).

## Vector construction and *C. trachomatis* transformation

### Constructs used in co-transfection experiments

Mammalian vectors expressing Cdu1-GFP constructs were kindly provided by Jonathan Pruneda (Oregon Health and Science University, OR). Briefly, geneblocks encoding full length Cdu1 (LGV L2 434 Bu, CTL0247) (amino acids 1–401), Cdu1 lacking its transmembrane domain (amino acids 71–401), and Cdu1 lacking its catalytic domain (amino acids 1–130) were generated and inserted into the pOPIN-GFP vector *Berrow et al., 2007* by In-Fusion cloning (Takara Bio; 638947), resulting in Cdu1 constructs with a C-terminal eGFP-His tag preceded by a 3 C protease cleavage site. The Flag-InaC mammalian expression vector was derived from a Gateway entry clone containing the *C. trachomatis* Serovar D/UW-3/CX CT813 (*inaC*) ORF (amino acids 41–264) obtained from a *C. trachomatis* ORFeome library (*Roan et al., 2006*). The entry vector was used as a donor plasmid for Gateway based transfer into a modified pcDNA DEST53 (Thermo Fisher Scientific; 12288015) vector in which the cycle 3 *GFP* ORF was removed. A NEB Q5-Site Directed Mutagenesis Kit (New England Biolabs; E0554S) was used to introduce a 3 X Flag epitope tag at the N-terminus of the CT813 ORF and a stop codon at the end of the CT813 ORF. L2 *ipaM* (CTL0476), L2 CTL0480, and L2 *cpoS* (CTL0481) ORFs were PCR amplified from cell lysates derived from Vero cells infected with wild type L2 LGV 434 Bu with primers containing attB sequences (primers *ipaM* forward, *ipaM* reverse, CTL0480 forward, CTL0480 reverse, *cpoS* forward, and *cpoS* reverse). Primer sequences can be found in *Supplementary file 10*. PCR amplicons were used as donors for Gateway BP Clonase based transfers into the donor vector pDONR221 (ThermoFisher Scientific; 12536017) to generate entry plasmids. Entry plasmids were used to transfer *ipaM*, CTL0480, and *cpoS* into the Gateway destination vector pcDNA3.1/nV5-DEST (ThermoFisher Scientific; 12290010) by Gateway LR Clonase based reactions. The resulting mammalian expression vectors express IpaM, CTL0480, and CpoS with V5-epitopes fused to their N-terminus.

### pBOMB4-MCI-based plasmids

CTL0480 and *ipaM* ORFs were amplified by PCR from cell extracts derived from Vero cells infected with wild type LGV L2 434 Bu. The CTL0480 ORF, 149 b.p of upstream sequence, and a 3 X FLAG epitope was amplified by PCR using primers RBP628 and RBP629. The CTL0476 (*ipaM*) ORF, 400 b.p of upstream sequence, and a 3 X FLAG epitope was amplified with primers RBP623 and RBP624. CTL0480 and *ipaM* amplicons were digested with Not1 and Pst1 and cloned into Not1 and Pst1 digested pBOMB4-MCI (*Bauler and Hackstadt, 2014*) to generate pBOMB4-MCI_CTL0480-3X Flag and pBOMB4-MCI_IpaM-3X Flag plasmids, respectively. pBOMB4-MCI_Cdu1-3XFlag plasmids were generated by PCR amplification of 175 b.p of genomic sequence directly upstream of the L2 434 Bu CTL0247 (*cdu1*) ORF and the entire *cdu1* ORF tagged with a C-terminal 3 X Flag epitope tag (primers RBP460 and RBP461). PCR amplicons were generated from gradient purified LGV L2 434 Bu

EBs and cloned into a pCR-Blunt II TOPO vector using a Zero Blunt TOPO PCR cloning Kit (Thermo Fisher Scientific; K2800J10) according to the manufacturer instructions. Cdu1 catalytic variants were generated with a NEB Q5-Site Directed Mutagenesis Kit (New England Biolabs; E0554S) using the *cdu1*p-*cdu1*-3X Flag construct cloned into pCR-Blunt II TOPO as a template and following the manufacturer instructions. Primers for introducing base pair changes were designed on the NEBaseChanger website. The Cdu1[C345A] variant was generated by changing the *TGC* codon located at positions 1033–1035 in the *cdu1* ORF to *GCT* (primers RBP525 and RBP526). The Cdu1[I225A] variant was generated by substituting the *ATC* codon located at positions 673–675 for *GCT* (primers RBP527 and RBP528). The Cdu1[K268E] variant was generated by introducing an *A802G* base pair substitution (primers RBP529 and RBP530). Wild type *cdu1*p-*cdu1*-3XFLAG and all three *cdu1* variants were digested with Not1 and Pst1 and ligated into Not1 and Pst1 digested pBOMB4-MCI (*Bauler and Hackstadt, 2014*). Primer sequences can be found in *Supplementary file 10*.

### p2TK2_SW2-inaC-3XFlag
The CTL0184 (*inaC*) ORF and 250 b.p of upstream sequence was amplified by PCR from cell extracts derived from Vero cells infected with L2 434 Bu and cloned into the p2TK2_SW2 vector (*Agaisse and Derré, 2013*). A NEB Q5-Site Directed Mutagenesis Kit (New England Biolabs; E0554S) was used to insert a 3 X FLAG epitope sequence at the C-terminus (stop codon removed) of the CTL0184 ORF to generate the p2TK2_SW2-InaC-3XFlag plasmid.

pBOMB4-MCI based plasmids and p2TK2_SW2-InaC-3X Flag plasmids were transformed into corresponding *Chlamydia* strains, and transformants were selected with 10 U/mL Penicillin G and plaque purified as previously described (*Kędzior and Bastidas, 2019*). All primer sequences and plasmids generated in this study are listed in *Supplementary file 10* and Key Resources Table.

## Subcellular fractionation
HeLa cells ($2.16 \times 10^7$ cells/strain) seeded in six well plates were mock infected or infected with *Chlamydia* L2 434 Bu strains. At indicated time points, cells were washed with ice-cold PBS, collected in ice-cold PBS with a cell scraper, and transferred to 15 mL conical tubes. Cell suspensions were centrifuged at 500 x *g* for 5 min at 4 °C and cell pellets were resuspended in 400 µL of ice-cold subcellular fractionation buffer (20 mM HEPES (pH 7.4), 10 mM KCl, 2 mM MgCl$_2$, 1 mM EDTA, 1 mM EGTA) supplemented with 1 mM DTT and a 1 x cOmplete Mini-EDTA free protease inhibitor cocktail (Sigma-Aldrich; 11836170001). Cells were incubated on ice for 20 min and lysed with 30 strokes of a Dounce homogenizer. Cell lysates were sequentially centrifuged twice at 720 x *g* for 5 min at 4 °C to remove intact nuclei. Supernatants were centrifuged at 10,000 x *g* for 5 min at 4 °C and the heavy membrane (inclusion) fraction was recovered and resuspended in IP lysis buffer (25 mM Tris-Hcl, 150 mM NaCl, 1% NP-40, 5% Glycerol) supplemented with 1 mM PMSF and a 1 x cOmplete Mini-EDTA free protease inhibitor cocktail (Sigma-Aldrich; 11836170001).

## Immunoprecipitations (IPs)
### Transfections and GFP-immunoprecipitations
HEK 293T cell monolayers ($1.32 \times 10^7$ cells/transfection) seeded in 10 cm cell culture dishes pre coated with poly-L-Lysine (Sigma-Aldrich; P4707) were grown to 50% confluency and transfected with 10 µg of each plasmid used per co-transfection in a 1.5–1 ratio of jetOPTIMUS (Polyplus; 101000051) transfection reagent to total plasmid DNA according to the manufacturer instructions. At 24 hpi, transfected cells were lysed in IP lysis buffer (described above) supplemented with 1 mM PMSF and 1 x cOmplete Mini-EDTA free protease inhibitor cocktail (Sigma-Aldrich; 11836170001). Lysates were transferred to Eppendorf tubes, sonicated, and cleared by centrifugation (21,000 x *g*, 15 min, 4 °C). Supernatants containing 2 mg of total protein were incubated with magnetic GFP-Trap agarose (Proteintech; gta) for 1 hr at 4 °C with rotation. Beads were washed according to manufacturer instructions and immunoprecipitated proteins eluted with 2 X Laemmli sample buffer.

### Flag immunoprecipitations
Mock and infected HeLa cell monolayers ($1.44 \times 10^7$ cells) grown in six well plates were lysed in IP lysis buffer (described above) and transferred to Eppendorf tubes. Lysates were sonicated and cleared by centrifugation (21,000 x *g*, 15 minutes, 4 °C). Supernatants containing 2 mg of total protein were

pre-cleared by incubating with Protein A/G PLUS-Agarose (Santa Cruz Biotechnology; sc-2203) for 30 min at 4 °C followed by sedimentation of agarose resins by centrifugation. Supernatants were incubated with M2-anti Flag mouse mAb (1:400) (Sigma-Aldrich; F1804) overnight at 4 °C with rotation followed by incubation with Protein A/G PLUS-Agarose for 3 hours at 4 °C with rotation. Agarose resins were sedimented and washed according to manufacturer instructions and immunoprecipitated proteins eluted with 50 µL of 100 µg/mL 3xFLAG peptides (APExBIO; A6001).

## Acetylated lysine immunoprecipitations

Heavy membrane (inclusion) subcellular fractions isolated from infected HeLa cells and containing 1 mg of total protein were pre-cleared by incubating with Protein A/G PLUS-Agarose (Santa Cruz Biotechnology; sc-2203) for 30 min at 4 °C followed by sedimentation of agarose resins by centrifugation. Supernatants were incubated with an anti-acetylated lysine rabbit antibody (Cell signaling; #9441) (1:100) overnight at 4 °C with rotation followed by incubation with Protein A/G PLUS-Agarose for 3 hr at 4 °C with rotation. Agarose resins were sedimented and washed according to manufacturer instructions and immunoprecipitated proteins eluted with 50 µL of 2 X Laemmli sample buffer.

Input (8 µg of total protein for Slc1 and alpha Tubulin blots, and 25 µg for all other blots) and immunoprecipitates (GFP, Flag, proteins acetylated at lysines) were loaded onto 4–15% Mini-PROTEAN and TGX Stain Free Protein Gels (BIO-RAD; 4568084), transferred to PVDF membranes (Bio-Rad; 1620177), blocked with 5% Milk/TBSt, and incubated with primary antibodies overnight at 4 °C. Protein signals were detected with Goat anti-mouse (H+L) IgG (Thermo Fisher scientific; 31430) or Goat anti rabbit (H+L) IgG HRP (Thermo Fisher Scientific; 31460) conjugated secondary antibodies (1:1000 in 5% Milk/TBSt) and SuperSignal West Femto HRP substrate (Thermo Fisher Scientific; 34096). Antibody-bound membranes were imaged with a LI-COR imaging system (LI-COR, Inc).

## Identification Cdu1 lysine acetylation and phosphorylation sites by LC-MS/MS

HeLa cell monolayers (8.64x10⁷ cells/strain) grown in six well plates were infected with a wild type L2 strain transformed with empty pBOMB4-MCI (*Bauler and Hackstadt, 2014*) plasmid or with a wild type L2 strain transformed with a pBOMB4-MCI_*cdu1*-3X Flag plasmid. At 24 hpi infected cells were lysed and Flag tagged Cdu1 was immunoprecipitated as described above. Flag eluates from 3 independent biological replicates were sent to the Proteomics and Metabolomics Shared Resource Facility at Duke University for quantitative LC-MS/MS analysis. Samples were spiked with undigested casein, reduced with 10 mM dithiothreitol, and alkylated with 20 mM iodoacetamide. Eluates were then supplemented with 1.2% phosphoric acid and S-Trap (Protifi) binding buffer (90% Methanol, 100 mM TEAB). Proteins were trapped on the S-Trap, digested with 20 ng/µL Trypsin (Trypsin Gold, Mass Spectrometry Grade) (Promega; V5280), and eluted with 50 mM TEAB, 0.2% FA, and 50% ACN/0.2% FA. Samples were lyophilized and resuspended in 1% TFA/2% acetonitrile containing 12.5 fmol/µL yeast alcohol dehydrogenase. Quantitative LC/MS/MS was performed using a nanoAcquitiy UPLC system (Waters Corp) coupled to Thermo Scientific Orbitrap Fusion Lumos high-resolution accurate mass tandem mass spectrometer via a nanoelectrospray ionization source. Data was analyzed with Proteome Discoverer 2.3 (Thermo Fisher Scientific.) and MS/MS data searched against the *Chlamydia trachomatis* LGV L2 434 Bu reference database (NCBI:txid47472). Cdu1-Flag MS/MS data was analyzed with Mascot software (Matrix Science) using Trypsin/P specificity for N-terminal acetylation, lysine acetylation, lysine Ub, and S/T/Y phosphorylation identification. Analysis identified multiple acetylated and phosphorylated Cdu1 peptides and no Cdu1 ubiquitinated peptides. Data was viewed in Scaffold with Scaffold PTM (Scaffold Software).

## Interferon gamma sensitivity assays

Ct sensitivities to interferon-gamma was assayed as previously described (*Walsh et al., 2022*). Briefly, A549 cells were seeded in black 96-well clear-bottomed plates (Corning). The next day, cells were stimulated with 0 U/mL or 100 U/mL interferon gamma (IFNγ; Millipore, IF005) in DMEM supplemented with L-tryptophan (100 µg/mL). After 20 hr, cells were infected in technical duplicate with indicated *Chlamydia* strains at an MOI of 2. At 24 hr post-infection, plates were fixed with cold 4% PFA in PBS for 20 min. Samples were nuclear stained with Hoechst in PBS for 10 min and sealed using an aluminum adhesive (Thermo Fisher). Inclusions and host cell nuclei were imaged and quantified using

the CellInsight CX5 High Content Screening platform (Thermo Fisher; CX51110). Relative bacterial infectivities were calculated as the number of inclusions divided by the total number of host nuclei for each sample. Interferon sensitivity was calculated by normalizing the infectivities of each strain to it's 'untreated' (-IFNγ) control and expressed as a percentage.

### Isolation and imaging of *Chlamydia* inclusion extrusions

HeLa cell monolayers (1.2x10⁶ cells) were infected with Ct strains at MOIs of 0.8. At 48 hpi, infected monolayers were imaged using an EVOS FL Cell Imaging System (ThermoFisher Scientific) equipped with a 20 x/0.4 NA objective and a CCD camera. Following imaging, growth media was removed, cell monolayers washed with fresh growth media, and monolayers incubated for an additional 4 hr at 37 °C. At 52 hpi growth supernatants were collected and transferred to Eppendorf tubes. Extrusions were enriched by centrifugation (1500 rpm, 5 min) and pellets (not always visible) containing extrusions were resuspended in 30 µL of 4% Formaldehyde/PBS supplemented with Hoechst (2 µg/mL) and 0.2% Trypan Blue Solution (Gibco, 0.4%). Extrusions were analyzed by plating 10 µL drops on a glass side (without coverslips) and immediately imaged using an EVOS FL Cell Imaging System (ThermoFisher Scientific) equipped with a 20 x/0.4 NA objective and a CCD camera. Intact extrusions were identified based on morphology, lacking nuclei, and being impermeable to trypan blue. Images were opened in ImageJ (*Schneider et al., 2012*) and enumeration of inclusions and extrusions was performed manually. The sizes of individual extrusions and inclusions were determined by manually tracing a line around the perimeter of each extrusion and inclusion in ImageJ and measuring perimeter length. All measurements were exported to Microsoft Excel. Data plots and statistical analyses were done with Prism 9 (GraphPad) software. Datasets were analyzed for significance using a paired student t-test.

### Image analysis

Line scan profiles of Cdu1 co-localization with inclusion membrane proteins was performed with ImageJ *Schneider et al., 2012* by tracing a line through regions of interest and plotting fluorescent signal intensities with the Plot Profile function. Localization of CTL0480, recruitment of MYPT1, and association of Ub with Ct inclusions was performed manually from maximum projections in ImageJ. Assessment of F-actin recruitment to Ct inclusions was performed manually in ImageJ by projecting four to five sections in order to capture the entire inclusion. Redistribution of Golgi around the Ct inclusion was measured in ImageJ from maximum projections. The perimeters of individual inclusions were manually traced, and lengths measured. The length of dispersed Golgi was measured by tracing and measuring the length of the GM130 signal directly adjacent to each inclusion. Dispersed Golgi length was divided by inclusion perimeter length. All measurements were exported to Microsoft Excel for quantification. Data plots and statistical analyses were done with Prism 9 (GraphPad).

### Quantification and statistical analysis

Quantifications were generated from three independent experiments and measurements derived from blinded images. Data plots and statistical analyses were done with Prism 9 (GraphPad) software. Datasets were analyzed for significance using a paired student t-test, one-way ANOVAs with a Student-Newman-Keuls post hoc test, or two-way ANOVAS with a Turkey post hoc test. Data graphs show means and error bars represent standard error. p-values less than 0.05 are defined as statistically significant. The indicated statistical test for each experiment can be found in the figure legends.

## Acknowledgements

We thank LifeSensors and the Duke Proteomics and Metabolomics Shared Resource Center for their proteomics services. We thank the Duke Light Microscopy Core Facility for microscopy services. We also thank Marcela Kokes for generating the IncA-Flag constructs used in this study. This work was supported by NIH grants GM142486 to JNP, AI103197 to JC, AI140019 to RJB and AI134891 to RHV.

## Additional information

### Funding

| Funder | Grant reference number | Author |
|---|---|---|
| National Institutes of Health | AI140019 | Robert J Bastidas |
| National Institutes of Health | AI134891 | Raphael H Valdivia |
| National Institutes of Health | AI103197 | Jorn Coers |
| National Institutes of Health | GM142486 | Jonathan N Pruneda |

The funders had no role in study design, data collection and interpretation, or the decision to submit the work for publication.

### Author contributions

Robert J Bastidas, Conceptualization, Data curation, Formal analysis, Supervision, Funding acquisition, Validation, Investigation, Visualization, Methodology, Writing - original draft, Project administration, Writing - review and editing; Mateusz Kędzior, Investigation, Methodology, Verified generation of the cdu1::GII mutant strain and showed interaction of Cdu1-GFP variants with InaC, IpaM, and CTL0480 in transfected HEK cells; Robert K Davidson, Investigation, Methodology, Quantified inclusion production and RNF213 localization in unprimed and IFNgamma-primed A549 cells; Stephen C Walsh, Investigation, Methodology, Quantified inclusion production and RNF213 localization in unprimed and IFNgamma-primed A549 cells; Lee Dolat, Resources, Generated the cdu1::GII strain; Barbara S Sixt, Jonathan N Pruneda, Jorn Coers, Resources; Raphael H Valdivia, Conceptualization, Data curation, Supervision, Funding acquisition, Writing - review and editing

### Author ORCIDs

Robert J Bastidas ⬤ http://orcid.org/0000-0001-9219-572X
Jonathan N Pruneda ⬤ http://orcid.org/0000-0002-0304-4418
Raphael H Valdivia ⬤ http://orcid.org/0000-0003-0961-073X

Reviewer #1 (Public Review): https://doi.org/10.7554/eLife.87386.3.sa1
Reviewer #2 (Public Review): https://doi.org/10.7554/eLife.87386.3.sa2
Reviewer #3 (Public Review): https://doi.org/10.7554/eLife.87386.3.sa3
Author Response https://doi.org/10.7554/eLife.87386.3.sa4

## Additional files

### Supplementary files

• Supplementary file 1. TUBE1 co-precipitating proteins mass spectrometry data for all biological replicates.

• Supplementary file 2. Numerical data used to generate all volcano plots.

• Supplementary file 3. Numerical mass spectrometry data used to identify high confidence TUBE1 human co-precipitating proteins.

• Supplementary file 4. Numerical data for Gene Ontology (GO) classification of TUBE1 human co-precipitating proteins.

• Supplementary file 5. Numerical mass spectrometry data used to identify high confidence TUBE1 *Chlamydia* co-precipitating proteins.

• Supplementary file 6. Human ubiquitinated peptides mass spectrometry data.

• Supplementary file 7. *Chlamydia* ubiquitinated peptides mass spectrometry data.

• Supplementary file 8. Numerical mass spectrometry data used tp identify high confidence ubiquitinated proteins.

• Supplementary file 9. Antibodies used in this study.
• Supplementary file 10. Primer sequences used in this study.
• MDAR checklist

## Data availability

Unprocessed (raw) proteomics data received from LifeSensors can be found in Supplementary Table 1. Original data used for microscopy and western blots in this study can be found at Mendeley data repository. Source data for each Figure contains numerical data used to generate the figures.

The following dataset was generated:

| Author(s) | Year | Dataset title | Dataset URL | Database and Identifier |
|---|---|---|---|---|
| Robert B, Mateusz K, Robert D, Stephen W, Lee D, Barbara S, Jonathan P, Jorn C, Raphael V | 2023 | The acetylase activity of Cdu1 regulates bacterial exit from infected cells by protecting Chlamydia effectors from degradation | https://doi.org/10.17632/xt3nmkm375.1 | Mendeley Data, 10.17632/xt3nmkm375.1 |

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

# Appendix 1

## Appendix 1—key resources table

| Reagent type (species) or resource | Designation | Source or reference | Identifiers | Additional information |
|---|---|---|---|---|
| Gene (*Chlamydia trachomatis* L2 434 Bu) *cdu1* | | BV-BRC (Bacterial and Viral Bioinformatics resource center); Refseq (NCBI Reference Sequence Database) | fig\|471472.4.peg.264 (BV-BRC) CTL0247 (Refseq) | |
| Gene (*Chlamydia trachomatis* L2 434 Bu) *inaC* | | BV-BRC; Refseq | fig\|471472.4.peg.199 (BV-BRC) CTL0184 (Refseq) | |
| Gene (*Chlamydia trachomatis* L2 434 Bu) *ipaM* | | BV-BRC; Refseq | fig\|471472.4.peg.511 (BV-BRC) CTL0476 (Refseq) | |
| Gene (*Chlamydia trachomatis* L2 434 Bu) CTL0480 | | BV-BRC; Refseq | fig\|471472.4.peg.516 (BV-BRC) CTL0480 (Refseq) | |
| Gene (*Chlamydia trachomatis* L2 434 Bu) *cpoS* | | BV-BRC; Refseq | fig\|471472.4.peg.517 (BV-BRC) CTL0481 (Refseq) | |
| Gene (*Chlamydia trachomatis* L2 434 Bu) *incA* | | BV-BRC; Refseq | fig\|471472.4.peg.406 (BV-BRC) CTL0374 (Refseq) | |
| Gene (*Chlamydia trachomatis* L2 434 Bu) *slc1* | | BV-BRC; Refseq | fig\|471472.4.peg.323 (BV-BRC) CTL0299 (Refseq) | |
| Gene (*Chlamydia trachomatis* L2 434 Bu) *rpoB* | | BV-BRC; Refseq | fig\|471472.4.peg.608 (BV-BRC) CTL0567 (Refseq) | |
| Gene (*Chlamydia trachomatis* L2 434 Bu) *garD* | | BV-BRC; Refseq | fig\|471472.4.peg.421 (BV-BRC) CTL0390 (Refseq) | |
| Gene (*Homo sapiens*) | *MYPT1* | NCBI | PPP1R12A | |
| Cell line (*Homo sapiens*) | Hela | ATCC | Cat# CCL-2 RRID:CVCL_0030 | |
| Cell line (*Cercopithecus aethiops*) | Vero | ATCC | Cat# CCL-81 RRID:CVCL_0059 | |
| Cell line (*Homo sapiens*) | HEK 293T | ATCC | Cat# CRL-3216 RRID:CVCL_0063 | |
| Cell line (*Homo sapiens*) | A549 | ATCC | Cat# CCL-185 RRID:CVCL_0023 | |
| Transfected construct (*Chlamydia trachomatis* L2 43 Bu) | pOPIN-GFP_Cdu1 FL (aa 1–401) | Jonathan Pruneda (Oregon Health and Science University) | N/A | Used for Cdu1 co-immunoprecipitations in transfected cells |
| Transfected construct (*Chlamydia trachomatis* L2 43 Bu) | pOPIN-GFP_Cdu1 TMD- (aa 71–401) | Jonathan Pruneda (Oregon Health and Science University) | N/A | Used for Cdu1 co-immunoprecipitations in transfected cells |
| Transfected construct (*Chlamydia trachomatis* L2 43 Bu) | pOPIN-GFP_Cdu1 CD- (aa 1–130) | Jonathan Pruneda (Oregon Health and Science University) | N/A | Used for Cdu1 co-immunoprecipitations in transfected cells |
| Transfected construct (*Chlamydia trachomatis* L2 43 Bu) | pCDNA-DEST53 (w/o GFP)_InaC (CT813)–3XFLAG | This paper | N/A | Construct for showing Cdu1-InaC co-IP in transfected cells |
| Transfected construct (*Chlamydia trachomatis* L2 43 Bu) | pcDNA3.1/nV5-DEST_IpaM | This paper | N/A | Construct for showing Cdu1-IpaM co-IP in transfected cells |

*Appendix 1 Continued on next page*

*Appendix 1 Continued*

| Reagent type (species) or resource | Designation | Source or reference | Identifiers | Additional information |
|---|---|---|---|---|
| Transfected construct (*Chlamydia trachomatis* L2 43 Bu) | pcDNA3.1/nV5-DEST_CTL0480 | This paper | N/A | Construct for showing Cdu1-CTL0480 co-IP in transfected cells |
| Transfected construct (*Chlamydia trachomatis* L2 43 Bu) | pcDNA3.1/nV5-DEST_CpoS | This paper | N/A | Construct for showing lack of Cdu1-Cpos co-IP in transfected cells |
| Biological sample (*Chlamydia trachomatis* L2 434 Bu) | *Chlamydia trachomatis* LGV biovar L2 434 Bu (L2) | Richard Stephens (UC Berkeley) | N/A | Reference *Chlamydia trachomatis* (L2) strain |
| Biological sample (*Chlamydia trachomatis* L2 434 Bu) | L2 pBOMB-MCI (Parent: LGV L2 434 Bu) | This paper | N/A | L2 strain transformed with empty expression plasmid |
| Biological sample (*Chlamydia trachomatis* L2 434 Bu) | L2 pBOMB-MCI_CTL0480-3XFlag (Parent: LGV L2 434 Bu) | This paper | N/A | L2 strain transformed with plasmid expressing CTL0480-3XFlag |
| Biological sample (*Chlamydia trachomatis* L2 434 Bu) | L2 pBOMB-MCI_IpaM-3XFlag (Parent: LGV L2 434 Bu) | This paper | N/A | L2 strain transformed with plasmid expressing IpaM-3XFlag |
| Biological sample (*Chlamydia trachomatis* L2 434 Bu) | L2 pBOMB-MCI_Cdu1-3XFlag (Parent: LGV L2 434 Bu) | This paper | N/A | L2 strain transformed with plasmid expressing Cdu1-3XFlag |
| Biological sample (*Chlamydia trachomatis* L2 434 Bu) | L2 pBOMB-MCI_Cdu1$^{C345A}$-3XFlag (Parent: LGV L2 434 Bu) | This paper | N/A | L2 strain transformed with plasmid expressing Cdu1$^{C345A}$–3XFlag |
| Biological sample (*Chlamydia trachomatis* L2 434 Bu) | L2 pBOMB-MCI_Cdu1$^{K268E}$-3XFlag (Parent: LGV L2 434 Bu) | This paper | N/A | L2 strain transformed with plasmid expressing Cdu1$^{K268E}$–3XFlag |
| Biological sample (*Chlamydia trachomatis* L2 434 Bu) | L2 *cdu1*::GII *aadA* (Parent: LGV L2 434 Bu) | This paper | N/A | L2 *cdu1* null strain (Spec$^R$) |
| Biological sample (*Chlamydia trachomatis* L2 434 Bu) | L2 *cdu1*::GII *aadA* pBOMB-MCI (Parent: L2 *cdu1*::GII *aadA*) | This paper | N/A | L2 *cdu1* null strain transformed with empty expression plasmid |
| Biological sample (*Chlamydia trachomatis* L2 434 Bu) | L2 *cdu1*::GII *aadA* pBOMB-MCI_Cdu1-3XFlag (Parent: L2 *cdu1*::GII *aadA*) | This paper | N/A | L2 *cdu1* null strain transformed with plasmid expressing WT Cdu1-3XFlag |
| Biological sample (*Chlamydia trachomatis* L2 434 Bu) | L2 *cdu1*::GII *aadA* pBOMB-MCI_Cdu1$^{C345A}$-3XFlag (Parent: L2 *cdu1*::GII *aadA*) | This paper | N/A | L2 *cdu1* null strain transformed with plasmid expressing WT Cdu1$^{C345A}$-3XFlag |
| Biological sample (*Chlamydia trachomatis* L2 434 Bu) | L2 *cdu1*::GII *aadA* pBOMB-MCI_Cdu1$^{I225A}$-3XFlag (Parent: L2 *cdu1*::GII *aadA*) | This paper | N/A | L2 *cdu1* null strain transformed with plasmid expressing WT Cdu1$^{I225A}$-3XFlag |
| Biological sample (*Chlamydia trachomatis* L2 434 Bu) | L2 *cdu1*::GII *aadA* pBOMB-MCI_Cdu1$^{K268E}$-3XFlag (Parent: L2 *cdu1*::GII *aadA*) | This paper | N/A | L2 *cdu1* null strain transformed with plasmid expressing WT Cdu1$^{K268E}$-3XFlag |
| Biological sample (*Chlamydia trachomatis* L2 434 Bu) | L2 Rif-R (Parent: L2 434 Bu) | PMID:22232666 | N/A | Rifampin resistant L2 strain (L2 434 Bu) |
| Biological sample (*Chlamydia trachomatis* L2 434 Bu) | M407 (*inaC C307T*, InaC Q103*) (Parent: L2 Rif-R) | PMID:25920978 | N/A | L2 strain bearing a nonsense mutation in *inaC* (*inaC* null srain) |
| Biological sample (*Chlamydia trachomatis* L2 434 Bu) | M407 p2TK2 (Parent: M407) | PMID:25920978 | N/A | L2 *inaC* null strain transformed with empty expression plasmid |
| Biological sample (*Chlamydia trachomatis* L2 434 Bu) | M407 p2TK2_InaC (Parent: M407) | PMID:25920978 | N/A | L2 *inaC* null strain transformed with plasmid expressing WT InaC (untagged) |
| Biological sample (*Chlamydia trachomatis* L2 434 Bu) | M407 p2TK2_InaC-3X Flag (Parent: M407) | This paper | N/A | L2 *inaC* null strain transformed with plasmid expressing WT InaC-3X Flag |
| Biological sample (*Chlamydia trachomatis* L2 434 Bu) | *inaC*::GII *bla* (Parent: L2 434 Bu) | PMID:28465429 | N/A | L2 *inaC* null strain (Penicillin$^R$) |

*Appendix 1 Continued on next page*

*Appendix 1 Continued*

| Reagent type (species) or resource | Designation | Source or reference | Identifiers | Additional information |
|---|---|---|---|---|
| Biological sample (*Chlamydia trachomatis* L2 434 Bu) | *ipaM*::GII *cat* (Parent: L2 434 Bu) | PMID:37530528 | N/A | L2 *ipaM* null strain (Chloramphenicol[R]) |
| Biological sample (*Chlamydia trachomatis* L2 434 Bu) | *ctl0480*::GII *aadA* (Parent: L2 434 Bu) | PMID:30555802 | N/A | L2 *ctl0480* null strain (Spec[R]) |
| Biological sample (*Chlamydia trachomatis* L2 434 Bu) | M923 (*incA* C589T, IncA R197*) (Parent: L2 Rif-R) | PMID:25920978 | N/A | L2 strain bearing a nonsense mutation in *incA* (*incA* null srain) |
| Biological sample (*Chlamydia trachomatis* L2 434 Bu) | M923 pBOMB-MCl (Parent: L2 M923) | PMID:28041929 | N/A | L2 *incA* null strain transformed with empty expression plasmid |
| Biological sample (*Chlamydia trachomatis* L2 434 Bu) | L2 434 Bu (Parent of L2 *garD*::GII) | PMID:36084633 | N/A | Wild type L2 parent of *garD* null strain |
| Biological sample (*Chlamydia trachomatis* L2 434 Bu) | L2 *ctl0390*::GII *aadA* (*garD*::GII) | PMID:36084633 | N/A | L2 *garD* null strain (Spec[R]) |
| Antibody | CT L2 Cdu1 (a.a 71–401) Rabbit polyclonal | This paper | N/A | Dilution IF: 1:100 Dilution WB 1:500 Antibody raised against L2 Cdu1 |
| Antibody | CT Cap1 Rabbit polyclonal | PMID:26981769 | N/A | Dilution IF: 1:250 |
| Antibody | CT CT813 (InaC) (D/UW-3/CX) Mouse monoclonal | PMID:16861671 | N/A | Dilution IF: 1:100 |
| Antibody | CT CTL0480 Rabbit polyclonal | PMID:23727243 | N/A | Dilution IF: 1:100 |
| Antibody | CT IncA Mouse monoclonal | Dan Rockey Oregon State Univ. Corvallis | N/A | Dilution IF: 1:100 |
| Antibody | CT IpaM Mouse monoclonal 12ES IgG1 | PMID:11207561 | N/A | Dilution IF: 1:100 Dilution WB 1:200 See *Supplementary file 9* |
| Antibody | CT RpoB Rabbit polyclonal | Ming Tang (UC Irvine) | N/A | Dilution WB 1:100 See *Supplementary file 9* |
| Antibody | CT Slc1 Rabbit polyclonal | PMID:24586162 | N/A | Dilution WB 1:400 See *Supplementary file 9* |
| Antibody | Hs Acetylated lysine Rabbit polyclonal | Cell signalling | Cat# 9441 RRID:AB_331805 | Dilution WB 1:4000 See *Supplementary file 9* |
| Antibody | Hs Alpha tubulin clone B-5-1-2 Mouse monoclonal | Sigma-Aldrich | Cat# T5168 RRID:AB_477579 | Dilution WB 1:500 See *Supplementary file 9* |
| Antibody | Flag epitope Mouse monoclonal | Sigma-Aldrich | Cat# F3165 RRID:AB_259529 | Dilution WB 1:1000 See *Supplementary file 9* |
| Antibody | Flag epitope Mouse monoclonal | Sigma-Aldrich | Cat# F1804 RRID:AB_262044 | Dilution IF: 1:250 |
| Antibody | Hs GM130 Mouse monoclonal | BD Biosciences | Cat# 610822 RRID:AB_398141 | Dilution IF: 1:1000 |
| Antibody | Lys48-linkage specific polyubiquitin (D9D5) Rabbit monoclonal | Cell signalling | Cat# 8081 RRID:AB_10859893 | Dilution WB 1:1000 See *Supplementary file 9* |
| Antibody | Hs MYPT1 Rabbit polyclonal | US Biological | Cat# M9925-01C RRID:AB_2927397 | Dilution IF: 1:100 |
| Antibody | Hs Ubiquitin (P4D1) Mouse monoclonal | Cell signalling | Cat# 3936 RRID:AB_331292 | Dilution IF: 1:50 |
| Antibody | V5 epitope Mouse monoclonal | Abcam | Cat# ab27671 RRID:AB_471093 | Dilution WB 1:5000 See *Supplementary file 9* |

*Appendix 1 Continued*

| Reagent type (species) or resource | Designation | Source or reference | Identifiers | Additional information |
|---|---|---|---|---|
| Antibody | Hs RNF213 Rabbit Polyclonal | Sigma | Cat# HPA003347 RRID:AB_1079204 | Dilution IF: 1:1000 |
| Antibody | Rabbit IgG-HRP Goat polyclonal | ThermoFisher | Cat# 31460 RRID:AB_228341 | Dilution WB 1:1000 See *Supplementary file 9* |
| Antibody | Mouse IgG-HRP Goat polyclonal | ThermoFisher | Cat# 31430 RRID:AB_228307 | Dilution WB 1:1000 See *Supplementary file 9* |
| Antibody | Rabbit IgG-A488 Goat polyclonal | ThermoFisher | Cat# A-11008 RRID:AB_143165 | Dilution IF: 1:1000 |
| Antibody | Rabbit IgG-A647 Goat polyclonal | ThermoFisher | Cat# A-21244 RRID:AB_2535812 | Dilution IF: 1:1000 |
| Antibody | Mouse IgG-A488 Goat polyclonal | ThermoFisher | Cat# A-11001 RRID:AB_2534069 | Dilution IF: 1:1000 |
| Antibody | Mouse IgG-A647 Goat polyclonal | ThermoFisher | Cat# A-21235 RRID:AB_2535804 | Dilution IF: 1:1000 |
| Sequence-based reagent | See *Supplementary file 10* | This paper | | Primers used in this study |
| Peptide, recombinant protein | 3xFLAG peptide | APExBIO | Cat# A6001 | Peptide for eluting Flag immunoprecipitated proteins |
| Peptide, recombinant protein | Recombinant human interferon gamma (IFNγ) | Millipore | Cat# IF005 | For inducing Type II interferon responses in cell cultures |
| Peptide, recombinant protein | LGV L2 434 Bu Cdu1 recombinant protein | Jonathan Pruneda (Oregon Health and Science University) | N/A | Recombinant Cdu1 protein used to generate Rabbit polyclonal antibodies against Cdu1 |
| Commercial assay or kit | TargeTron gene knockout system | Sigma-Aldrich | Cat# TA0100 | Kit for generating *cdu1*::GII null strain |
| Chemical compound, drug | PR-619 (DUB inhibitor) | Sigma-Aldrich | Cat# SML0430 | Used during TUBE affinity enrichment |
| Chemical compound, drug | MG132 | Sigma-Aldrich | Cat# 474791 | Proteasome inhibitor |
| Chemical compound, drug | TUBE-1 | LifeSensors | Cat# UM401M | Pan-polyubiquitination affinity capture reagent |
| Chemical compound, drug | Acti-stain 488 (Phalloidin 488) | Cytoskeleton Inc. | Cat# PHDG1 | Reagent for visualizing Actin by IF |
| Chemical compound, drug | ChromoTek GFP-Trap Agarose | Proteintech | Cat# gta | Agarose for immunoprecipitating GFP tagged proteins |
| Chemical compound, drug | Protein A/G PLUS-Agarose | Santa Cruz Biotechnology | Cat# sc-2203 | Agarose for immunoprecipitations |
| Software | Targetronics | Targetronics, LLC; https://www.targetrons.com/ | | Algorithm for generating TargeTron homing sequences |
| Software | Proteome Discoverer 2.3 | Thermo Fisher; https://www.thermofisher.com/in/en/home/industrial/mass-spectrometry/liquid-chromatography-mass-spectrometry-lc-ms/lc-ms-software/multi-omics-data-analysis/proteome-discoverer-software.html | | Proteomics software |
| Software | Mascot software | Matrix Science; https://www.matrixscience.com | | Proteomics software |
| Software | MaxQuant 1.6.2.3 | https://www.maxquant.org | | Proteomics software |
| Software | Scaffold PTM | Proteome Software; https://www.proteomesoftware.com/products/scaffold-ptm | | Software for viewing proteomics data |

*Appendix 1 Continued on next page*

*Appendix 1 Continued*

| Reagent type (species) or resource | Designation | Source or reference | Identifiers | Additional information |
|---|---|---|---|---|
| Software | VolcaNoseR | PMID:33239692; https://huygens.science.uva.nl/VolcaNoseR/ | | Online software for generating volcano plots |
| Software | Metascape | PMID:30944313; https://metascape.org/gp/index.htm#/main/step1 | | Online program for Gene Ontology clustering |
| Software | DAVID Bioinformatic Resources | PMID:19033363; PMID:19131956 https://david.ncifcrf.gov/tools.jsp | | Online program for Gene Ontology clustering |
| Software | Image J | PMID:22930834; https://imagej.nih.gov/ij/ | | Software for image viewing and processing |
| Software | NEBaseChanger | New England Biolabs; https://nebasechanger.neb.com | | Cloning website |
| Software | Prism 9 | GraphPad; https://www.graphpad.com/updates/prism-900-release-notes | | Software for statistical analysis |
| Software | HCS Studio Cell Analysis Software | Thermo Fischer Scientific | Cat# CX51110 | Software for high content image analysis |

