## [Editor Report · eLife assessment]

This **important** study combines state-of-the art proteomics and genetic manipulation of *Chlamydia trachomatis* to study the function of a chlamydial effector, Cdu1, with deubiquitination and acetylation activities. **Solid** evidence is provided to show that Cdu1 is able to protect itself and three other chlamydial effectors, which are involved in the control of chlamydial egress from host cells, from ubiquitin-mediated degradation, and that this depends on the acetylation activity of Cdu1, but not on its deubiquitination activity. This work will be of interest to microbiologists and cell biologists studying host cell-pathogen interactions.

---

## [Referee Report · Reviewer #1 (Public Review)]

The objective of this study was to investigate the influence of the *C. trachomatis* effector Cdu1 on the ubiquitination of proteins in infected host cells and its correlation with the previously identified role of Cdu1 in facilitating Golgi distribution around the Chlamydia inclusion.

To achieve this, the authors created a cdu1-null mutant in *C. trachomatis* and employed proteomics to analyze ubiquitinated proteins in cells infected with Cdu1-producing and Cdu1-deficient chlamydiae, comparing them to mock-infected cells. The results revealed that, among the proteins specifically ubiquitinated after infection with Cdu1-deficient chlamydiae, three were other *C. trachomatis* effectors (InaC, IpaM, and CTL0480), members of a large family of Chlamydia effectors (Incs) that insert in the inclusion membrane.

Subsequently, the authors focused on understanding how Cdu1 shields InaC, IpaM, and CTL0480 from ubiquitination and the implications of this protection for the protein levels and functions of these Incs during infection. Data is presented showing that Cdu1 can bind to InaC, IpaM, and CTL0480, and protects these Incs and itself from ubiquitination and proteasomal degradation. This protective role of Cdu1 is dependent on its acetylation, but not on its deubiquitinating activity. Host cells infected by the cdu1 null mutant displayed defects resembling those observed in cells infected by inaC, ipaM, or ctl0480 null mutants.

Additionally, it was previously established that CTL0480 inhibits a chlamydial egress pathway involving the extrusion of the inclusion. This study now revealed that InaC and IpaM also play a role in promoting the extrusion of *C. trachomatis* inclusion, and the cdu1 null mutant exhibited a defect in this process. This leads to the title's conclusion that Cdu1 regulates chlamydial exit from host cells by safeguarding specific *C. trachomatis* effectors from degradation.

In summary, this work is excellent and impressive, both technically and conceptually, providing mechanistic insights into the action of Cdu1. The data provides convincing support for the proposed model, illustrating how the acetylation activity of Cdu1 protects itself and three Incs (InaC, IpaM, and CTL0480) from degradation. While the study indicates that the observed phenotypes in cells infected by the cdu1 null mutant are linked to reduced levels of InaC, IpaM, and CTL0480, these Incs are still detectable in cells infected by the cdu1 null mutant. Even if very unlikely, this leaves room for the possibility that Cdu1 directly promotes assembly of F-actin and Golgi repositioning around the inclusion, MYPT1 recruitment to the inclusion, and extrusion of the inclusion. Nevertheless, the major significance of this work lies in the integration of proteomics and chlamydial genetics to unveil a unique mechanism in which one effector controls the levels of other effectors, emphasizing the intricate relationships among bacterial effectors injected into host cells.

---

## [Referee Report · Reviewer #2 (Public Review)]

Based on the corresponding author's response, the questions I raised were not addressed for various reasons. This is not necessarily a negative. The authors indicated that most of the points raised will be addressed in a separate manuscript. Specifically, the Cdu1 targeting of IkBa. They mentioned intriguing findings regarding IkBa in cells infected with a cdu1-null strain *C. trachomatis* in their response to reviewers. Similar to this, there appears to be a planned manuscript that will address the question of the timing of CTL0480's function in inclusion extrusion.

The lack of more direct infection-related evidence of Cdu1 interaction with various type III effectors was raised; and the authors attributed this to technical difficulties and low abundance of starting materials. It was not clear if they tried other approaches to demonstrate interaction.

Another suggestion was the quantitation of the three target effectors of Cdu1 in wild type and cdu1-null background. The authors provided western blot data and immunofluorescence images that revealed potential differences in stability/turnover kinetics. The authors might want to discuss the implications of the different kinetics of stability/turnover. For example, if all three proteins are necessary for optimal extrusion of inclusions, and concertedly act to mediate this process, all three would need to be present at the required levels. Could this be a temporal regulation strategy? Does acetylation also regulate function, interactions, etc.?

In short, the response to some of the questions is forthcoming in the form of follow-up manuscripts. New observations on the different stability profiles could be elaborated in the Discussion section, with a brief discussion on functional and/or regulatory implications.

---

## [Referee Report · Reviewer #3 (Public Review)]

In this article by Bastidas et al. the authors examine the functions of the Chlamydia deubiquitinating enzyme 1 (Cdu1) during infections of human cells. First, a mutant lacking Cdu1 but not Cdu2 was constructed using targetron and quantitative proteomics was used to identify differences in ubiquitinated proteins (both host and bacterial) during infection. While they found minimal changes in host protein ubiquitination, they identified three Chlamydia effector proteins, IpaM, InaC and CTL0480 were all ubiquitinated in the absence of Cdu1. Microscopy and immunoprecipitations found Cdu1 directly interacts with these Chlamydia effectors and confirmed that Cdu1 mediates the stabilization of these effectors at the inclusion membrane during late infection time points. Surprisingly rather than deubiquitination driving this stabilization, the acetylation function of Cdu1 was required, and acetylation on lysine residues prevented degradative ubiquitination of Cdu1, IpaM, InaC and CTL0480. In line with this observation the authors show that loss of Cdu1 phenocopies the loss of single effector mutants of InaC, IpaM and CTL0480, including golgi stack formation and the recruitment of MYPT1 to the inclusion. The aggregation of changes to the Chlamydia inclusion does not alter growth but controls extrusion of chlamydia from cells with reduced extrusion in Cdu1 mutant Chlamydia infections. The strengths of the manuscript are the range of assays used to convincingly examine the biochemical and cellular biology underlying Cdu1 functions. The finding that acetylation of lysine residues is a mechanisms for bacterial effectors to block degradative ubiqutination is impactful and will open new investigations into this mechanism for many intracellular pathogens. The authors revisions to the manuscript have addressed my primary concerns and the authors present compelling arguments for remaining questions that are outside the scope of this study. Altogether this is an important series of findings that help to understand the mechanisms underpinning Chlamydia pathogenesis using orthologous methods and is an impactful study.

---

## [Author Response]

The following is the authors’ response to the original reviews.

**RESPONSE TO REVIEWERS:**

**Reviewer #1 (Recommendations For The Authors):**
I think the manuscript of this excellent work can be improved, especially in writing (including a suggestion in the title) and presentation (Figure 6); Also some additional specific experiments and analyses could be important, as I suggest below,1. For the title, perhaps a shorter "The acetylase activity of Cdu1 protects Chlamydia effectors from degradation" would be better to convey the major significance of this work. Of course, Cdu1 must regulate the function of InaC, IpaM and CTL0480. But perhaps it is speculative to think that egress is the major function of these effectors as their activity on other host cell processes during the cycle could eventually impact the extrusion process indirectly.

Although we concur with the insights provided by reviewer 1, we wish to underscore that a significant breakthrough presented in our study revolves around the regulation of Chlamydia exit by Cdu1. Consequently, we believe that this noteworthy discovery should be incorporated into the title.

2. For the writing:a. The description of ubiquitination and DUBs could be synthesized to the essential, so that space is gained to explain things that then come a bit out of the blue in the results (what are Incs, the specific functions of InaC, IpaM, and CTL0480 - at least place the citations in lines 110-112 next to the corresponding Incs -, Cdu2, etc - see specifics below)

In lines 182-196 of the revised manuscript, we have incorporated additional contextual information concerning the roles of Incs, along with descriptions of the functions of InaC, IpaM, and CTL0480.

b. In the Results, there is a lot of Chlamydia- and maybe lab-specific jargon that could be significantly simplified for the more general reader. I detail some suggestions below in the specific issues.

We have improved the readability of our manuscript for a general audience by removing Chlamydia-specific terminology from the entire text and figures.

3. For the figures:a. Figure 6, this figure could be reorganized: why two graphs in panel D? If detailed quantifications were done, perhaps in panel B just zoom on the examples of Golgi distributed/compacted? And again the labelling Rif-R L2, L2 pBOMB, M407 p2TK2, etc, simplify?

Figure 6 has undergone restructuring. The representative images have been relocated to Supplemental Figures 5 and 6, while we have introduced sample images demonstrating F-actin assembly and Golgi repositioning. Furthermore, the quantification of Golgi dispersal has been streamlined into a single panel. Additionally, we have simplified the labeling of the strains utilized in the study.

b. Figure 3, in the labelling, WT, inaC null, cdu1::GII wouldn't be enough? Leave the details to the legend and/or M&M.

We have simplified the labeling of Ct strains in Figure 3.

c. Figure 3C, these arrowheads should not be so symmetric (small arrows instead?) and it is unclear that the indicated cells do not show CTL0480.

We have substituted arrowheads with small arrow symbols and have also revised the Figure to incorporate a new representative image that prominently illustrates the absence of CTL0480 at the inclusion membrane of some cdu1::GII inclusions within infected Hela cells at 36 hpi.

4. Experiments:a. In Figure 7, at least extrusion should be analysed also with the Cdu1-deficient strain expressing Ac-deficient Cdu1 and the inaC and ipaM phenotypes should be complemented.

We have conducted additional experiments to analyze extrusion production in Hela cells infected with a cdu1 null strain expressing the acetylase-deficient Cdu1 variant. We have incorporated the relevant data into revised Figure 7, where the impact of this strain on extrusion production and size is presented. Additionally, we updated Supplemental Figure 8 to include data illustrating the number of inclusions produced by this strain. We have also addressed these new results in the revised manuscript (lines 424-432). We are currently complementing inaC and ipaM mutant strains with various InaC and IpaM constructs that will be used in a follow up manuscript.

b. Does overexpression of InaC, IpaM, or CTL0480 in a cdu1-null background prevent the degradation of these Incs and suppress the defects of cells infected by the cdu1 mutant (F-actin, Golgi, MYPT1)? This would show that the multiple phenotypes displayed by cells infected by the cdu1 null mutant are indeed related to the decreased levels of InaC, IpaM and CTL0480.

We opted not to include data from the overexpression of these effectors in a cdu1-null background due to an unexpected decrease in shuttle plasmid load during overexpression. This development prompted concerns regarding the potential detrimental effects of overexpressing these effectors in the absence of Cdu1. Data supporting this observation are not included in this report.

c. Figures 3A and 3B should be quantified (it says it is from 3 independent experiments). It would be important to have a relative perspective of how much Cdu1 protects these Incs over time (for InaC, it would also be nice to have the 36 and 48 hpi time-point). This is in contrast with the microscopy data in Figure 5, which illustrates very clear effects, and the quantification is a bit redundant.

In Figure 3, we have incorporated a new Western Blot image showing endogenous InaC protein levels in Hela cells following infection with both WT Ct and cdu1::GII strains at 24, 36, and 48 hours post-infection (hpi). Additionally, we have quantified the Western Blot signals for both InaC and IpaM, and these results are also presented in Figure 3. The quantification of MYPT1 recruitment has been relocated to a supplementary figure. We have also included details regarding the methodology employed for the quantification of Western Blot signals in the Materials and Methods section.

d. What is the subcellular localization of InaC, IpaM, CTL0480 and Cdu1 when analysed by transfection? Does Cdu1 bind to of InaC, IpaM, CTL0480 in infected cells? If this was attempted and unsuccessful it should be mentioned.

In transfected HEK cells, InaC, IpaM, CTL0480, and Cdu1 all exhibit cytoplasmic localization with a diffuse pattern (data not shown). Despite our efforts, we encountered challenges in observing co-immunoprecipitation of Cdu1 with all three Incs in infected Hela cells at 24 hpi, We have duly acknowledged this limitation in our findings, as reflected in line 221-226 of the revised manuscript.

5. Specific issues:Line 87, "propagule" is really needed to describe the EB?

The EB is the infectious form of Chlamydia species that spreads within the host to renew its life cycle; thus, "propagule" is a suitable term to characterize the EB.

Exocytosis implies fusion with the plasma membrane so "inclusion is exocytosed" (line 91) is not entirely correct.

In line 91 of the revised manuscript, we referred to extrusion as the exit of an intact inclusion from the host cell and omitted the use of "exocytosed" to describe this process.

Line 126, "a Ct L2 (LGV L2 434 Bu) background". Maybe "a Ct cdu1-null strain" would be enough and leave the detail for Materials and Methods.

In line 128 of the revised manuscript, we omitted "(LGV L2 434 Bu)" to avoid using jargon that may be unfamiliar to readers not well-versed in Chlamydia terminology.

Line 138, in the previous Pruneda et al, Nature Microbiol 2018, the title of figure 4 is "ChlaDUB deubiquitinase activity is required for *C. trachomatis* Golgi fragmentation", so why raise this hypothesis? And why in the end is the acetylation activity of Cdu1 that promotes Golgi distribution? I think this related with infection vs transfection experiments but it deserved to be briefly explained/discussed.

In lines 140-142 of the revised manuscript, we provide clarification that the DUB activity of Cdu1 is required for Golgi fragmentation in transfected cells. This observation supports our initial hypothesis suggesting that the DUB activity of Cdu1 is also required for Golgi distribution in infected cells, and our rationale for identifying targets of its DUB activity.

Lines 147-155, what is the relevance of this non-ubiquitinated proteins that come along? Couldn't this be synthesized?

We have included a discussion on non-ubiquitinated proteins, as they could potentially encompass proteins that interact with those protected by Cdu1. This perspective provides supplementary insights into the roles of proteins targeted for ubiquitination in the absence of Cdu1. The results of this analysis have been succinctly summarized in a single paragraph within the initial manuscript (lines 151-159 of the revised manuscript).

Line 170, I think it is the first time that "Type 3 secretion"; perhaps explain in the introduction.

Type 3 secretion systems have been extensively characterized and discussed in the literature, and we anticipate that the majority of our readers are well-acquainted with this secretory mechanism.

Line 184, I think it is the first time "microdomains" are mentioned; perhaps mention in the introduction.

The definition of "microdomains" has been provided in line 191 of the revised manuscript.

Figure 2, as it stands the analysis with truncated Cdu1 proteins adds little to the work. Binding to the Incs seems to be affected when the TM domain is not present, but it still binds. And this is in a transfection context.

The results depicted in Figure 2, involving truncated Cdu1 proteins, illustrates that Cdu1 is capable of interacting with InaC, IpaM, and CTL0480 even in the absence of infection. This finding serves as evidence suggesting that all three Incs could potentially serve as direct targets for Cdu1 activity. As a result, we prefer to keep these findings in the manuscript.

Line 219, "late stages of infection", this is shown (albeit not completely quantified) for IpaM and CTL0480, but not for InaC.

In the revised Figure 3, we show InaC protein levels at 24, 36, and 48 hours post-infection, and we have incorporated quantitative data for both InaC and IpaM protein levels in the context of Hela cells infected with both WT L2 and cdu1::GII strains. This updated figure serves to emphasize the pivotal role of Cdu1 in safeguarding all three Incs during the late stages of infection.

Line 233, "pBOMB-MCI backbone" - is this needed in the Results section? And this refers to Figure 4 while pBOMB appear already in Fig. 3.

We have removed “pBOMB-MCI backbone” in the revised manuscript.

Line 236, should be cdu1 endogenous promoter.

In line 265 of the revised manuscript we have replaced Cdu1 with cdu1 (italicized).

Line 263, WT.

In line 293 of the revised manuscript we replaced “wild type” with “WT”.

Line 277, IncA instead of "the Inc protein IncA".

In the manuscript we wanted to emphasize that IncA is also an inclusion membrane protein, therefore we have included “the Inc protein IncA” in the revised manuscript to avoid any confusion.

How does the data in Figure 5 relates to the relatively few proteins ubiquitinated in cells infected with cdu1-mutant Ct? These Ub-labelling corresponds to ubiquitinated InaC, IpaM and CTL0480?

The findings presented in Figure 5 demonstrate that the acetylase activity of Cdu1 plays a crucial role in enabling Ct to block all ubiquitination events taking place on or in proximity to the periphery of the inclusion membrane. This encompasses Cdu1 targets that might not have been identified through our proteomic analysis.

Lines 299-301, "M923 inclusions", there is certainly a clear way to write this.

In lines 326-327 and 332-332 of the revised manuscript, we have clarified that “M923” is an incA null strain to provide clarification.

Line 309, is "peripheries" correct?

We have changed “peripheries” with “periphery” in the revised manuscript (line 360).

Line 312, "Rif-R L2" and "M407" - can this be simplified?

In the revised manuscript, "Rif-R L2" was substituted with "WT L2" in lines 363 and 382, while "M407" was exchanged with "an inaC null strain" in lines 311, 367, and 368. These same replacements were applied to the Figures and their corresponding legends for consistency.

Lines 308-321, and 326-335, these % are all approximate figures and this should be made clear.

In lines 364-395 of the revised manuscript we have stated that all percentages are approximate values.

Fig. S1, kb and not k.b; what's the "+ control"; and is not really possible to have a PCR that works for the *? 3 kb is not that long.

In the updated Figure S1, we have corrected "k.b" to "kb". In the legend of Figure S1, we have clarified that the + control corresponds to the cdu2 locus. Moreover, we could not cleanly amplify a 3 kb PCR product from bacteria in whole cell lysates of infected mammalian cells (Vero cells).

Fig. S2, kb and not k.b, bp and not b.p

In the updated Figure S2, we have corrected “k.b” with “kb” and “b.p” with “bp”.

**Reviewer #2 (Recommendations For The Authors):**
Figure 1 describes an affinity-based purification and mass spectrometric identification of differentially ubiquitinated proteins (host and chlamydial). Through different permutations of combinations of infection (mock, wild type, and Cdu1 mutant), three effectors, IpaM, InaC, and CTL0480, were identified as putative targets of Cdu1. The authors used a high-stringency cutoff, which could explain identification of only three targets. Having said this, the localization of Cdu1 to the inclusion membrane would be expected to also narrow down the number of targets. Interestingly, Cdu2, another deubiquitinase remained active in these experiments, which could have affected identification of Cdu1 targets. The authors addressed this issue by referring to previously reported structural studies. A somewhat glaring omission is the lack of reference to NF-kB as a substrate of ChlaDub1/Cdu1. In experiments by Le Negrate et al., ChlaDub1 ectopic overexpression in cells led to the deubiquitination of IkB-alpha, thus inhibiting the nuclear translation of NF-kB. Based on the inclusion membrane localization of Cdu1 during infection, is the identification of IkB an artifact of overexpression of Cdu1, or is it still a bona fide Cdu1 target?

We conducted experiments using our cdu1 null strain to investigate whether IκBα could be a target of Cdu1 activity. While our findings are intriguing and relevant, it is not feasible to determine, at this stage, whether our findings result from a direct or indirect consequence of Cdu1 localizing to the inclusion membrane. Consequently, these findings extend beyond the scope of the current manuscript. We plan to explore the implications of our observations more deeply in a subsequent manuscript, where we intend to provide a more comprehensive and mechanistic analysis based on these preliminary findings. Additionally, we have referenced the potential targeting of IκBα by Cdu1 in lines 100-101 and 166-171 of the revised manuscript.

Figure 2 demonstrates the individual interaction of the identified effectors with Cdu1. Interaction at the inclusion membrane is inferred from colocalization studies, while protein-protein interaction is monitored using ectopic overexpression of tagged versions of Cdu1 and the individual effectors. This is somewhat of a weakness of the manuscript because the mechanism of action of Cdu1 towards its target hinges on protein-protein interaction.

Despite our efforts, we encountered challenges in co-immunoprecipitating endogenous Cdu1 with all three Incs in infected Hela cells at 24 hpi. There are multiple technical reasons as to why these interactions, which are predicted to be transient, will not be captured by bulk affinity approaches such as immunoprecipitations, especially when the starting materials are present in very low abundance. We acknowledged these limitations in our findings, as reflected in lines 221-226 of the revised manuscript.

Figure 3 provides the first evidence in this paper of the importance of the inferred interaction of Cdu1 with the three effectors. The authors show that the loss of cdu1 has stability consequences on the three effectors. This figure would benefit from quantifying InaC- or IpaM-positive inclusions in the same manner done with CTL0480. The timepoint-dependent effect of Cdu1 loss of function is intriguing. Do InaC and IpaM retention at the inclusion show the same timepoint-dependent characteristic?

In the revised Figure 3, we have incorporated InaC protein levels at 24, 36, and 48 hours post-infection. Additionally, we have included quantitative data representing both InaC and IpaM protein levels in HeLa cells infected with both WT L2 and cdu1::GII strains. The quantification of CTL0480 localization to cdu1::GII inclusions has been moved to a supplementary figure.

This updated figure illustrates that the absence of Cdu1 has a time-dependent impact on both InaC and IpaM. However, it is noteworthy that the kinetics of degradation for these two proteins diverge significantly.

For Figure 7, the authors should consider monitoring timing of inclusion extrusion to gain additional insight into the functional interactions between the effectors. For example, the loss of CTL0480 leads to increased extrusion, implying a role in delaying or suppressing extrusion. In a time-course experiment, a CTL0480 mutant could exhibit an earlier occurrence of inclusion extrusion.

One of the principal discoveries of this study is that Cdu1, InaC, IpaM, and CTL0480 collaborate to facilitate optimal extrusion of Ct from host cells. These findings represent a significant contribution to our understanding of how Chlamydia controls its exit from infected cells. We are currently in the process of expanding on these results. A forthcoming follow-up manuscript will provide more detailed and comprehensive exploration of these findings.

**Reviewer #3 (Recommendations For The Authors):**
Specific comments.a. I have some concerns related to the time point chosen for mass spec analysis and potential caveats and alternative interpretations. This work was done relatively early (24 hours) compared to the most convincing Cdu1 functions that occur later, thus this may limit the authors global understanding of protein changes. For example, the known substrate of Cdu1, Mcl-1 was not identified but this is altered relatively late during infection. Thus, the surprise that minimal host proteins are altered in ubiquitination may be partially driven by the timing of the assay. This should be more clearly discussed as a caveat.

In the revised manuscript (lines 166-171), we have acknowledged that there might be additional targets of Cdu1 that remain unidentified, primarily due to the specific time point we utilized in our study.

b. Another caveat to these studies is while the loss of Cdu1 alters different effectors stability and function and extrusion size, these changes do not modulate bacterial growth in cells. The authors speculate that regulating extrusion size may alter interactions with innate cells to drive dissemination. However, a previous study found defects in an animal model using a Cdu1 transposon mutant found decreased bacterial load in the genital tract. It is also possible that redundancy of effectors may mask importance in growth of Cdu1, but the authors strongly argue against redundancy of Cdu1 and Cdu2 so this weakens the authors argument here. These concepts and published data should be more directly discussed in the context of the authors proposed extrusion model and the role in driving Chlamydia growth and pathogenesis.

In our revised manuscript (lines 460-466) we propose that while we do not observe any growth impairments during Ct growth in the absence of Cdu1 in HeLa cells, the reduction in bacterial loads observed in murine models of infection with an independent cdu1 mutant strain (cdu1::Tn) may potentially be linked to defects in extrusion production or alterations in Cdu1-dependent regulation of extrusion size.

c. Recent studies have found that IFNg activation can result in dramatic changes in ubiquitination to pathogen containing vacuoles. While some of these are blocked by the newly found GarD, it seems possible that Cdu1 may also play a role (and perhaps use its deubiquinating activity) to further protect the inclusion. In light of published results showing that Cdu1 mutants have lower IFU burst size only in IFNg activated cells, this may be an important caveat in the current studies. This should be more directly addressed in the current manuscript.

We have incorporated two experimental findings indicating that the presence of Cdu1 is not required for Ct to defend itself against IFNγ cellular immunity in human cells. These recent discoveries are now presented in the updated Figure 5 and detailed in lines 338-355 of the revised manuscript.

d. On lines 433-434 the authors claim that Cdu1 is atypical since it is not encoded with the metaeffector/target pairs. However, this is an oversimplification of what is known about metaeffectors. For example, there are meta-effector/effector pairs that are not encoded together in Legionella (see table 1 DOI: https://doi.org/10.3390/pathogens10020108). Thus, the discussion should be adjusted. It seems Cdu1 is the first meta-effector found in Chlamydia, and maybe this should be highlighted more strongly rather than its uniqueness in this aspect of meta-effector/effector functions.

In lines 488-489 of the revised manuscript, we have removed the assertion that Cdu1 functions as an atypical metaeffector and emphasized that it represents the initial discovery of a metaeffector within Ct.